# SciLitLLM: How to Adapt LLMs for Scientific Literature Understanding

**Sihang Li**[1*†]**, Jin Huang**[2*†]**, Jiaxi Zhuang**[2†]**, Yaorui Shi**[1†]**, Xiaochen Cai**[2]**,**
**Mingjun Xu**[2†]**, Xiang Wang**[1‡]**, Linfeng Zhang**[2]**, Guolin Ke**[2]**, Hengxing Cai**[2‡]

[1]University of Science and Technology of China, [2]DP Technology

{sihang0520, xiangwang1223}@gmail.com, {huangjin, caihengxing}@dp.tech
[*]Equal contribution. [†]Work done while interning at DP Technology. [‡]Corresponding author.

## Abstract

Scientific literature understanding is crucial for extracting targeted information and garnering insights, thereby significantly advancing scientific discovery. Despite the remarkable success of Large Language Models (LLMs), they face challenges in scientific literature understanding, primarily due to (1) a lack of scientific knowledge and (2) unfamiliarity with specialized scientific tasks. To develop an LLM specialized in scientific literature understanding, we propose a hybrid strategy that integrates continual pre-training (CPT) and supervised fine-tuning (SFT), to simultaneously infuse scientific domain knowledge and enhance instruction-following capabilities for domain-specific tasks. In this process, we identify two key challenges: (1) constructing high-quality CPT corpora, and (2) generating diverse SFT instructions. We address these challenges through a meticulous pipeline, including PDF text extraction, parsing content error correction, quality filtering, and synthetic instruction creation. Applying this strategy, we present a suite of LLMs: **SciLitLLM**, specialized in scientific literature understanding. These models demonstrate promising performance on scientific literature understanding benchmarks. Our contributions are threefold: (1) We present an effective framework that integrates CPT and SFT to adapt LLMs to scientific literature understanding, which can also be easily adapted to other domains. (2) We propose an LLM-based synthesis method to generate diverse and high-quality scientific instructions, resulting in a new instruction set – **SciLitIns** – for less-represented scientific domains. (3) SciLitLLM achieves promising performance in scientific literature understanding benchmarks. We release the data processing codes[1] and model weights[2].

## 1 Introduction

Scientific literature understanding involves the systematic evaluation and interpretation of scientific texts and publications, to identify trends, extract targeted information, and garner insights (AI4Science & Quantum, 2023; Zheng et al., 2023), significantly contributing to scientific discovery. Concurrently, Large Language Models (LLMs) (OpenAI, 2023; Qwen, 2024; Touvron et al., 2023) have achieved remarkable success in natural language processing, prompting the development of domain-specific LLMs across various fields (Cui et al., 2024; Wu et al., 2023; Clusmann et al., 2023). However, recent studies (Wadden et al., 2024; Cai et al., 2024; Singh et al., 2023) indicate that LLMs face challenges when specializing in scientific literature understanding, particularly in context understanding and question answering. Take Figure 2 as

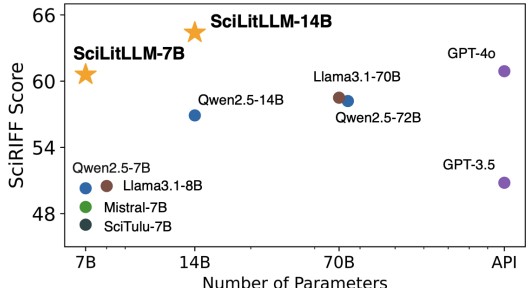

Figure 1: Average scores on SciRIFF of models with varying parameter sizes.

[1]https://github.com/dptech-corp/Uni-SMART
[2]https://huggingface.co/collections/Uni-SMART/scilitllm15-67283353ada975ba995629ef

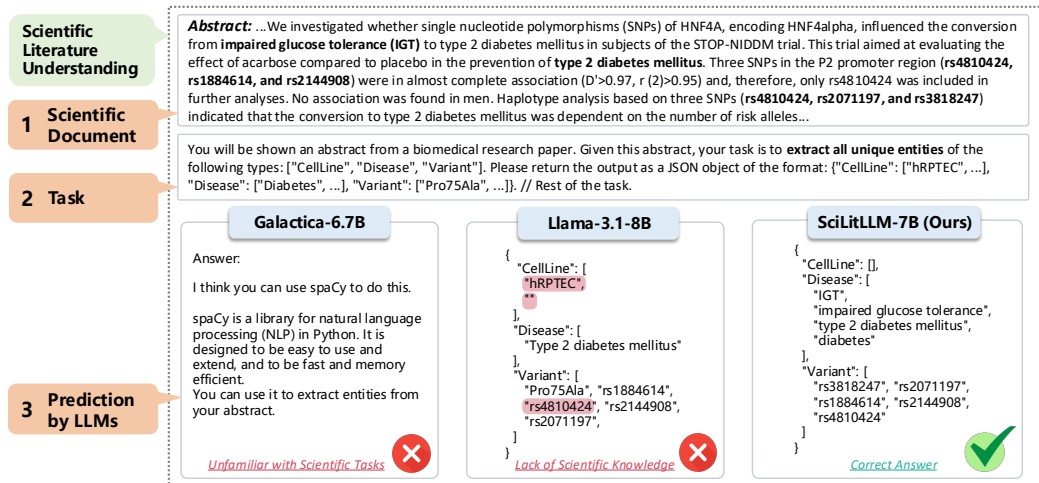

Figure 2: An example of scientific literature understanding in SciRIFF. It involves extracting accurate entities from a biomedicine paper. SciLitLLM-7B demonstrates sufficient scientific knowledge and instruction-following ability to accurately identify and extract these entities.

an example, where the LLM is asked to understand the content of a biomedical research paper and then extract the targeted information. LLMs' potential might be hindered by two major barriers: (1) a lack of *scientific knowledge*, which results in errors such as the missing important entities in Llama-3.1-8B (AI@Meta, 2024), and (2) unfamiliarity with *scientific tasks*, leading to the inability of Galactica-6.7B (Taylor et al., 2022) to follow task instructions accurately.

To make LLMs specialized in science-relevant tasks, existing studies mostly adopt two strategies, as illustrated in Figure 3: (1) Fine-tuning with scientific instructions (Wadden et al., 2024; Zhang et al., 2024b; Singhal et al., 2022). A general-purpose LLM is fine-tuned with collected domain-specific instructions to adapt it to science-relevant tasks. However, instruction fine-tuning alone is insufficient to imbue the models with comprehensive scientific knowledge. (2) Pre-training on scientific corpora (Beltagy et al., 2019; Zeng et al., 2022; Taylor et al., 2022). This approach involves training models on vast scientific corpora. While this method equips LLMs with domain knowledge, the lack of instruction-tuning confines them to solving relevant tasks. Moreover, it is hampered by substantial computational costs and data requirements (Ling et al., 2023; Yang et al., 2023). To address these obstacles while balancing efficiency, we propose a hybrid strategy that incorporates continual pre-training (CPT) and supervised fine-tuning (SFT), to simultaneously infuse domain knowledge and enhance domain-specific instruction-following capabilities.

However, as illustrated in Figure 4, developing a scientific literature understanding model using this CPT and SFT pipeline presents two critical requirements: **(1) High-quality CPT Corpora.** Scientific corpora, predominantly in PDF format such as textbooks and research papers, are not directly digestible for LLM training. Converting these documents to text with PDF parsing tools introduces formatting and syntax errors, degrading corpus quality. Worse still, scientific documents often contain segments that contribute little information (*e.g.*, references), necessitating quality control to filter them out. See the first row in Figure 4 for a comparison of high- and low-quality CPT texts. **(2) Diverse Scientific Instructions.** Effective instruction following for scientific literature understanding requires a large, high-quality, and diverse set of task-related instructions. However, to the best of our knowledge, there is a scarcity of well-designed instruction datasets for scientific literature understanding, and hiring human annotators to curate such a dataset from scratch is prohibitively expensive (Erdmann et al., 2019; Qiu et al., 2023). See the second row in Figure 4 for an illustration of high- and low-quality instructions.

To address these challenges, we devise an effective pipeline to construct high-quality domain corpora for CPT and diverse scientific instructions for SFT, as illustrated in Figure 5:

In the **CPT** stage for domain knowledge injection, we start with an extensive in-house corpus consisting of 73k textbooks and 625k academic papers in the scientific field, all in PDF format. Initially,

Figure 3: Comparison of strategies to adapt LLMs to scientific tasks.

Figure 4: Examples of high and low-quality CPT text and SFT instructions.

we leverage PyPDF2[3], a widely used open-source PDF parsing tool, to extract raw texts from these documents. We then employ a moderate yet powerful model, Llama3-8B-Instruct, to correct the format and spelling errors introduced by PDF parsing (*cf.* Section 2.1.1). Subsequently, we train a small text quality classifier to score the corpus and filter out texts of low educational value[4] in the scientific field (*cf.* Section 2.1.2). These two simple yet effective textual refinement and quality control measures ensure the high quality of our CPT corpus, culminating 12.7 billion tokens for CPT via the Qwen2.5 tokenizer.

In the **SFT** stage for domain instruction fine-tuning, to overcome the scarcity of domain-specific instructions and the high cost of human annotations, we propose a novel instruction synthesis method (*cf.* Section 2.2.1). It enables us to generate diverse instructions to better equip the model for domain-specific tasks. Moreover, we sequentially apply instruction duplication based on Levenshtein distance and an LLM-based filtering method to ensure the quality of synthetic instructions (*cf.* Section 2.2.2).

Having established such high-quality datasets, we apply the CPT-SFT integration strategy on a general-purpose LLM – Qwen2.5 (Qwen, 2024) and obtain SciLitLLM of two scales: 7B and 14B. Evaluations on benchmarks of scientific literature understanding demonstrate the effectiveness of our strategy. We observe promising performance enhancements, with an average improvement of 4.0% on SciAssess (Cai et al., 2024) and 10.1% on SciRIFF (Wadden et al., 2024), compared to the leading LLMs under 10B parameters. Notably, SciLitLLM-7B even outperforms Llama3.1 and Qwen2.5 with 70B parameters on SciRIFF. Additionally, SciLitLLM-14B achieves leading results on both benchmarks, surpassing other open-source LLMs. Further ablation studies demonstrate the effectiveness of each module in our pipeline.

In summary, our contributions are threefold: (1) We devise an effective and comprehensive pipeline to adapt general LLMs to a specific domain – scientific literature understanding. It combines continual pre-training (CPT) and supervised fine-tuning (SFT), to enhance scientific knowledge base and instruction-following capabilities for specialized domain tasks. (2) We propose a novel domain instruction synthesis method to curate instructions for scientific literature understanding, resulting in a new dataset – SciLitIns. (3) SciLitLLM, trained through the proposed pipeline, outperforms leading open-source LLMs on scientific literature understanding.

## 2 METHOD

In this section, we present the details of our proposed pipeline (*cf.* Figure 5): continual pre-training for scientific knowledge injection and supervised fine-tuning for scientific tasks enhancement. We refer to Appendix A for a comprehensive literature review on knowledge injection, domain adaptation, and scientific literature understanding for LLMs.

### 2.1 CPT FOR SCIENTIFIC KNOWLEDGE INJECTION

---

[3]https://pypdf2.readthedocs.io

[4]Phi models (Gunasekar et al., 2023; Li et al., 2023; Abdin et al., 2024) propose to determine the quality of a pre-training text by its educational value for a student whose goal is to learn basic domain concepts.

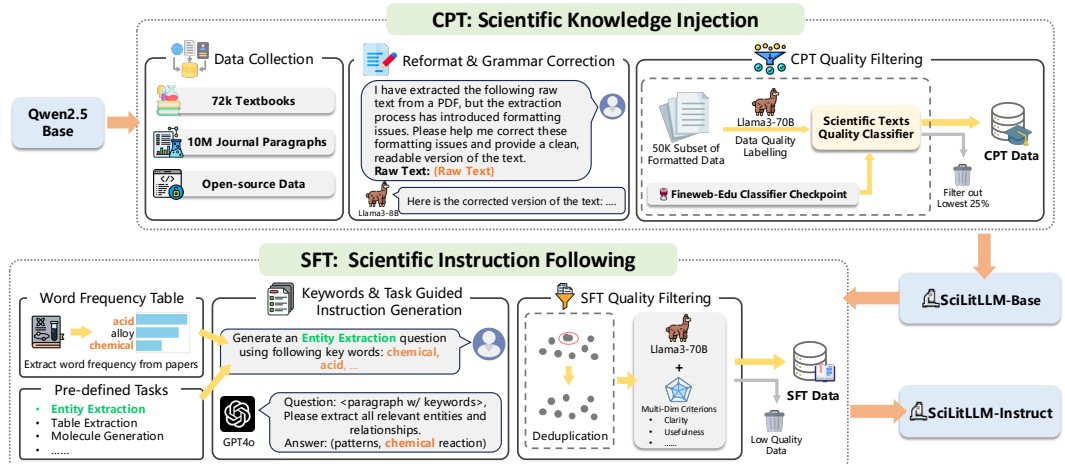

Figure 5: The pipeline of SciLitLLM consists of two key stages: continual pre-training (CPT) for scientific knowledge injection and supervised fine-tuning (SFT) for scientific instruction following. Specifically, the CPT stage involves several modules: PDF parsing, format & grammar correction (*cf.* Section 2.1.1), and quality filtering (*cf.* Section 2.1.2) modules. These modules ensure the model is equipped with high-quality scientific domain knowledge. The SFT stage includes LLM-based instruction generation (*cf.* Section 2.2.1) and instruction quality control (*cf.* Section 2.2.2) measures. These steps are designed to fine-tune the model's ability to follow scientific instructions accurately and effectively.

What are high-quality pre-training corpora? Researchers (Gunasekar et al., 2023; Li et al., 2023; Abdin et al., 2024) suggest that language models benefit from corpora that possess the same qualities as an exemplary textbook for human learners: clarity, self-containment, instructiveness, and balance. Recognizing the wealth of high-quality scientific textbooks and research papers over the past decades, we have curated a substantial collection of over 73,000 textbooks and 625,000 research papers within the scientific domain, ensuring all documents are copyright-compliant.

| Stage | Data source | Domain | #Doc/#Ins | # Tokens |
|---|---|---|---|---|
| CPT | In-house Textbooks | Science | 73k | 10B |
| | In-house Journals | Science | 625k | 2.7B |
| | Redpajama (Computer, 2023) | General | - | 11B |
| SFT | SciLitIns | Science | 93k | 86M |
| | SciRIFF (Wadden et al., 2024) | Science | 70k | 40M |
| | Infinity-Instruct | General | 3M | 1.7B |

Table 1: Data statistics of CPT and SFT stages. Underlined datasets are curated by us.

However, we still face two practical obstacles when dealing with those textbooks and research papers: (1) *Formatting and syntax errors.* Most textbooks and research paper documents are in PDF format, which is not directly digestible by LLMs. Converting these documents using tools like PyPDF2 often introduces formatting and syntax errors, which degrade the quality of the corpus. (2) *Corpus quality control.* Despite their overall high quality, textbooks and research papers also contain segments with little useful information, such as references and garbled text introduced during the PDF parsing process. To tackle these obstacles, we devised the following modules: format & grammar correction and CPT quality filter.

### 2.1.1 FORMAT & GRAMMAR CORRECTION

A parsed text from a PDF document often contains many formatting and syntax errors. To address this issue, we prompt Llama3-8B-Instruct, to correct these errors introduced during the PDF parsing process. Utilizing the vLLM (Kwon et al., 2023) backend, Llama3-8B-Instruct can process approximately 2.52 million tokens per Nvidia A100 GPU hour. The process takes over 5,000 A100 GPU hours to handle all the textbooks and research papers. Example texts – both before and after processing – along with the prompt template are provided in Appendix B to demonstrate the improvements through this correction process.

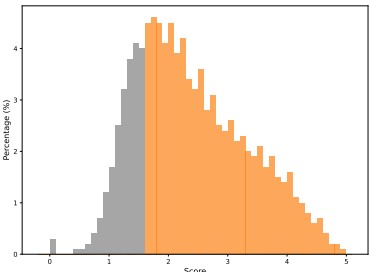

Figure 6: Score distribution of the CPT Data. The filtered-out 25% data are marked **gray**, while the remaining 75% are marked **orange**.

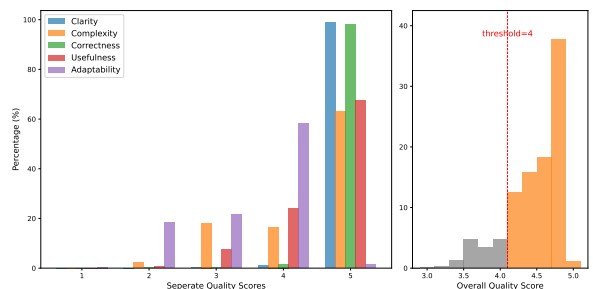

Figure 7: The quality of SciLitIns is evaluated from five aspects: clarity, complexity, correctness, usefulness, and adaptability. Instructions with an average score of less than 4 are filtered out.

### 2.1.2 CPT QUALITY FILTER

During CPT, maintaining the quality of the training corpus is crucial for effective knowledge injection. Given the extremely large scale of pre-training corpora, assessing quality through human annotation is not feasible (Erdmann et al., 2019; Qiu et al., 2023). Consequently, leading LLMs (*e.g.,* Phi (Gunasekar et al., 2023), Llama (Touvron et al., 2023), and Qwen2.5 (Qwen, 2024)) employ model-based quality filters. The typical process involves using larger LLMs to score the quality of a subset of texts, which then serve as labels for training small classifiers (*e.g.,* random forest (Gunasekar et al., 2023) and Bert (AI@Meta, 2024)) to annotate the entire training corpus. Inspired by this approach, we design a resource-efficient method based on a lightweight text quality classifier.

We first annotate a random subset of 50k CPT texts using Llama3-70B-Instruct. We adapt the quality assessment prompt from fineweb-edu-classifier (Anton et al., 2024), a widely-used quality classifier for web data, to evaluate the educational value (Gunasekar et al., 2023) of the scientific knowledge in each sampled text, assigning scores ranging from 0 (lowest quality) to 5 (highest quality). After annotation, we perform supervised transfer learning on the fineweb-edu-classifier checkpoint.

This process results in a scientific text quality classifier tailored for scientific corpus assessment. We then utilize this classifier to assess the quality of the entire CPT dataset and exclude the lowest-scoring 25% (See Figure 4 for concrete samples). The distribution of the scores is illustrated in Figure 6. We refer to Appendix C for more details about classifier training and sensitivity analysis of the filtering threshold.

### 2.1.3 CPT TRAINING SETTINGS

After preparing all the corpus, we perform CPT on Qwen2.5-Base (Qwen, 2024) for one epoch, encompassing 23.7 billion tokens (*cf.* Table 1), with a sequence length of 2,048 tokens. To maintain the model's general knowledge, we also include a similar scale of general corpus tokens from Redpajama (Computer, 2023). To stabilize the learning procedure, we gradually decrease the learning rate from $1 \times 10^{-5}$ to 0 with a cosine scheduler. To address overfitting, we apply a weight decay of 0.1 and gradients were clipped at a maximum value of 1.0. The CPT training took approximately 3 days on 32 Nvidia A100 GPUs for SciLitLLM-7B-Base and about 7 days for the 14B model.

## 2.2 SFT FOR SCIENTIFIC INSTRUCTION FOLLOWING

After performing CPT on an extensive scientific corpus to incorporate domain knowledge, we subsequently conduct SFT on domain-specific instructions to enhance the model's ability to understand scientific literature. We identify two major challenges in SFT for scientific instruction following:

- Existing instruction-tuning datasets in the scientific domain (Feng et al., 2024; Fang et al., 2023) primarily focus on fields such as physics, chemistry, and biology. Manually collecting instruction-tuning data for other less-represented vertical domains (*e.g.,* biomedicine, and material) is both time-consuming and costly (Erdmann et al., 2019; Qiu et al., 2023).

- Few instruction-tuning datasets adequately reflect the scenario of scientific literature understanding, which typically involves a segment of scientific literature accompanied by a question that requires deriving an answer from the text.

To address these challenges, we draw inspiration from leading models (*e.g.,* Nemotron-4 (Adler et al., 2024), Phi (Gunasekar et al., 2023), and Qwen (Yang et al., 2024)), which leverage existing LLMs to construct synthetic instruction sets. We propose a novel instruction synthesis method to curate instructions specifically for scientific literature understanding.

### 2.2.1 INSTRUCTION SYNTHESIS OF LESS-REPRESENTED DOMAINS

Unlike typical question-answer pairs, an instruction for a scientific literature understanding task comprises three components (Wadden et al., 2024): (1) a segment of scientific literature, (2) a question pertaining to the context, and (3) the corresponding answer. Simply prompting an LLM to generate a scientific context along with an associated question-answer pair – without variations in the instructions or parameters – often yields similar or repeated contents. Thus, we design a simple yet effective three-step pipeline to generate diverse and high-quality scientific contexts and corresponding question-answer pairs, consisting of the following:

1. *Probability table of domain keywords.* For a target scientific domain (*e.g.,* biomedicine, and material), we collect dozens of high-impact research papers via Google Scholar[5] and count the word frequency appearing in these papers. Then, we obtain a probability table of domain keywords.

2. *Scientific task list.* Since LLMs are expected to handle various types of scientific tasks, an instruction set with task diversity is essential. Therefore, we compile a list of task descriptions by including representative tasks from existing scientific NLP datasets (Wadden et al., 2024; Cai et al., 2024; Feng et al., 2024), covering as many scenarios as possible that an LLM may encounter in real applications.

3. *Instruction Generation.* Given a word probability table and the task list for a specific scientific domain, we sample 20 keywords and a task description each time. Subsequently, GPT-4o (OpenAI, 2023) is prompted to generate a scientific context containing the sampled keywords and a question-answer pair according to the provided task description.

Utilizing this pipeline, we obtain over 100k synthetic instructions covering less-represented scientific domains and various types of specialized tasks (details are presented in Appendix D.1).

### 2.2.2 INSTRUCTION QUALITY CONTROL

To ensure the diversity and quality of generated instructions, effective measures for quality control are essential. Specifically, we incorporate heuristic deduplication and LLM-based filtering.

1. *Heuristic deduplication.* Despite the measures taken during the generation process to prevent high homogeneity in the instructions, the generated data points may still contain similar questions or identical answers. To eliminate such redundancy, we implement a deduplication process to remove similar data points. See Appendix D.2 for details.

2. *LLM-based filtering.* Inspired by recent efforts (Eldan & Li, 2023; Chiang & Lee, 2023; Zhang et al., 2024c) to measure the quality of generated content using LLMs, we leverage Llama-3-70B-Instruct to assess the quality of generated instructions for five aspects: clarity, complexity, correctness, usefulness, and adaptability. We show the quality statistics of synthetic instructions in Figure 7. Instructions with an average score of less than 4 are filtered out. The detailed recipe for instruction quality evaluation with concrete examples is included in Appendix D.3.

Through instruction synthesis and quality control pipeline, we obtain **SciLitIns**, consisting of 93,894 high-quality and diverse instructions for scientific literature understanding.

### 2.2.3 SFT TRAINING SETTINGS

Our SFT training dataset consists of three parts: SciLitIns, SciRIFF (Wadden et al., 2024) and Infinity-Instruct[6], as shown in Table 1. Infinity-Instruct is a collection of more than twenty open-source instructions datasets, covering various general domains. SciRIFF and SciLitIns contain specialized instructions for scientific literature understanding. For both scales, we train for one epoch

---

[5] https://scholar.google.com/
[6] https://huggingface.co/datasets/BAAI/Infinity-Instruct

| Models | MMLU-Pro-Bio | MMLU-Pro-Chem | MMLU-Pro-Heal | MaScQA | Avg. |
|---|---|---|---|---|---|
| **# Parameters < 10B** | | | | | |
| Llama3.1-8B-Base | 56.35 | 25.88 | 44.50 | 53.28 | 45.00 |
| Qwen2.5-7B-Base | 68.36 | 48.37 | 51.64 | 56.23 | 56.15 |
| **SciLitLLM-7B-Base** | **70.45** | **51.21** | **54.06** | **59.27** | **58.75** |
| **# Parameters > 10B** | | | | | |
| Qwen2.5-14B-Base | 79.78 | 62.73 | 62.22 | 66.92 | 67.91 |
| **SciLitLLM-14B-Base** | **80.62** | **64.44** | **64.24** | **68.06** | **69.34** |

Table 2: Accuracy(%) comparison of base models. SciLitLLM outperforms Llama3.1 and Qwen2.5 with similar scales, demonstrating improved domain-specific understanding through continual pre-training on scientific corpora. Underline indicates those trained on domain-specific corpora.

| Dataset | Domain/ Task | # Parameter < 10B | | | | | # Parameter > 10B | | | | API | |
|---|---|---|---|---|---|---|---|---|---|---|---|---|
| | | Mistral-7B | Llama3.1-8B | Qwen2.5-7B | SciTulu-7B | SciLitLLM-7B | Qwen2.5-14B | Llama3.1-70B | Qwen2.5-72B | SciLitLLM-14B | GPT3.5 | GPT4o |
| SciAssess | Biology | 52.0 | 63.4 | 60.9 | 45.3 | **65.3** | 65.9 | **69.4** | 67.2 | 67.0 | 55.4 | 68.9 |
| | Chemistry | 34.0 | 46.1 | 47.9 | 19.0 | **55.4** | 60.0 | 62.5 | **63.7** | 63.4 | 37.3 | 68.0 |
| | Material | 36.5 | 50.9 | 48.9 | 31.3 | **53.7** | 57.9 | 59.2 | 59.4 | **62.6** | 37.0 | 62.0 |
| | Medicine | 25.1 | 30.2 | 28.3 | 19.9 | **32.4** | 31.5 | 37.4 | 37.8 | **39.9** | 31.8 | 45.8 |
| | Mean | 36.9 | 47.7 | 46.5 | 28.9 | **51.7** | 53.8 | 57.1 | 57.0 | **58.2** | 40.4 | 61.2 |
| SciRIFF | BioASQ | 43.2 | 45.0 | 45.3 | 37.5 | **51.2** | 48.0 | 43.5 | 46.7 | **53.9** | 47.3 | 46.7 |
| | BioR | 48.9 | 48.1 | 43.9 | 55.7 | **76.6** | 56.8 | 61.5 | 57.8 | **83.3** | 53.9 | 61.0 |
| | DiscMT | 44.5 | 68.8 | **73.7** | 61.5 | 71.1 | 70.5 | 77.3 | 54.7 | **77.7** | 67.9 | 78.3 |
| | EI | 17.2 | 17.2 | 23.3 | 11.6 | **23.5** | 27.2 | 26.0 | 26.3 | **30.8** | 19.2 | 24.7 |
| | MC | 47.5 | 49.7 | 50.8 | 34.6 | **70.7** | 54.2 | 60.9 | 58.0 | **72.1** | 47.8 | 58.7 |
| | MuP | **87.5** | 87.0 | 84.9 | 72.1 | 67.5 | 91.5 | 90.9 | **94.0** | 67.7 | 76.8 | 86.9 |
| | Qasper | 48.8/43.2 | 46.5/42.7 | 38.1/25.7 | **54.2**/38.6 | 50.7/**54.1** | 48.4/46.8 | 49.1/40.2 | **61.9**/50.5 | 54.0/**55.1** | 54.7/39.8 | 67.8/50.5 |
| | SciERC | 31.1 | 32.9 | 30.1 | 35.6 | **49.9** | 33.7 | 36.2 | 34.1 | **54.7** | 28.6 | 42.2 |
| | SciFact | 68.8/53.8 | 63.8/53.7 | 77.8/60.3 | 66.0/49.2 | **83.5/67.3** | 83.1/65.6 | 87.3/**70.3** | 86.5/69.6 | **91.0**/68.6 | 69.7/53.3 | 84.3/68.7 |
| | Mean | 48.6 | 50.5 | 50.3 | 47.0 | **60.6** | 56.9 | 58.5 | 58.2 | **64.4** | 50.8 | 60.9 |

Table 3: Model performances on scientific literature understanding benchmarks: SciAssess and SciRIFF. SciLitLLM-7B and SciLitLLM-14B achieve leading performance compared with open-source models of similar scales. The best-performing models smaller than 10B and larger than 10B are highlighted in bold. Results for SciTulu-7B, GPT-3.5, and GPT-4o on SciRIFF are taken from its original papers, while all other results are produced by us. Underline indicates those trained on domain-specific instructions.

on Infinity-Instruct to cultivate their general instruction-following abilities, then for five epochs on SciLitIns and SciRIFF for scientific literature understanding enhancement. The training is conducted with a sequence length of 4,096, a maximum learning rate of $1 \times 10^{-5}$, and a cosine scheduler. The SFT training takes approximately 32 hours for the 7B and 70 hours for the 14B model on 32 A100 GPUs, resulting in SciLitLLM-7B-Instruct and SciLitLLM-14B-Instruct.

## 3 EXPERIMENTS

In this section, we perform experiments to answer the following research questions: (Q1) How does SciLitLLM perform on scientific literature understanding tasks? (Q2) Can CPT with domain-specific corpora aid in scientific knowledge injection? (Q3) Can SFT with SciLitIns improve performance on scientific literature understanding tasks?

### 3.1 EXPERIMENTAL SETUP

#### 3.1.1 BENCHMARKS

To evaluate the performance of LLMs regarding scientific knowledge base and specialized task-solving abilities, our benchmarks include:

- *Base model benchmarks.* To assess their scientific knowledge, which serves as the foundation for scientific literature understanding, we evaluate the base models on scientific subjects – biology, chemistry, and health in MMLU-Pro (Wang et al., 2024) and MaScQA (Zaki et al., 2023) – a question answering dataset of materials science.

- *Instruction model benchmarks.* We evaluate the instruct models on scientific literature understanding benchmarks: SciRIFF (Wadden et al., 2024) and SciAssess (Cai et al., 2024). Due to the space limitation, we refer to their original paper for the detailed metrics of each task. Brief descriptions of them are provided in Appendix E.

### 3.1.2 BASELINES

- *Base model baselines.* We compare SciLitLLM-base against leading open-source base models: Llama-3.1 (AI@Meta, 2024), Qwen-2.5 (Qwen, 2024).

- *Instruction model baselines.* We benchmark leading instruction LLMs including GPT-[3.5, 4o] (Brown et al., 2020; OpenAI, 2023), Llama3.1-[8B, 70B] (AI@Meta, 2024) and Qwen2.5-[7B, 14B, 72B] (Yang et al., 2024; Qwen, 2024) and Mistral-7B (Jiang et al., 2023). We also report the performance of SciTulu-7B (Wadden et al., 2024), which is a fine-tuned Llama2-7B (Touvron et al., 2023) model on SciRIFF.

## 3.2 PERFORMANCE OVERVIEW (Q1)

### 3.2.1 BASE MODEL PERFORMANCE

The performance comparison of base models is shown in Table 2. SciLitLLM-base consistently outperforms other general base models across four scientific benchmarks, indicating its extensive domain knowledge. For instance, SciLitLLM-7B-Base achieves an average accuracy improvement of 2.60% over Qwen2.5-7B-Base, despite both models having a comparable parameter count. This improvement can be attributed to the specialized pretraining on high-quality domain-specific corpora. It benefits from exposure to scientific language patterns, terminology, and reasoning.

### 3.2.2 INSTRUCT MODEL PERFORMANCE

As shown in Table 3, compared with models with similar scale, SciLitLLM-7B achieves the highest performance in all 4 domains on SciAssess, outperforming the second-best model (*i.e.,* Llama3.1-8B) by a notable 4.0%. On SciRIFF, it surpasses baseline models by an even more substantial margin of 10.1%. These results highlight the effectiveness of our pipeline in adapting a general-purpose LLM into a specialized model for scientific literature understanding.

Specifically, one key contributor is our synthetic instruction dataset — SciLitIns, which offers comprehensive task coverage across diverse domains. It ensures that the model can generalize across tasks with varying input data and reasoning complexities, such as table extraction and molecule generation. For example, SciLitLLM-7B leads in 15 out of 27 sub-tasks on SciAssess, indicating its strong ability to adapt to the specialized requirements of these tasks.

Notably, SciLitLLM-14B shows significant performance gains over Qwen2-14B on both benchmarks, with a 4.4% improvement on SciAssess and a 7.5% improvement on SciRIFF. These results suggest that our approach effectively leverages the larger capacity of the 14B parameter model, enabling it to encode more nuanced scientific reasoning and domain-specific knowledge. Interestingly, SciLitLLM-14B also outperforms much larger open-source models, such as Llama3.1-70B and Qwen2.5-72B, despite having only one-fifth the number of parameters. This demonstrates that our continual pretraining and instruction fine-tuning pipeline is not merely scale-dependent but strategically optimized for domain adaptation.

The performance on SciRIFF further illustrates this optimization. Comparing with open-source models with similar scale, both SciLitLLM-7B and SciLitLLM-14B rank first in 8 out of 11 evaluations, including challenging tasks like Qasper and SciFact. These tasks require advanced abilities in scientific question answering and fact verification, respectively, which are likely bolstered by the diversity of high-quality examples in SciLitIns.

These findings underscore the broader importance of designing domain-specific LLMs. While general-purpose models, such as Llama and Qwen, perform well on broader tasks, their lack of domain specialization limits their effectiveness in fields like scientific literature understanding. The results demonstrate that the adaptation pipeline designed by us can yield promising performance without relying solely on increased model size.

## 3.3 ABLATION STUDY (Q2 & Q3)

We conducted ablation experiments on three key components in our pipeline: the CPT stage, the SFT data recipe, and the instruction quality filter, to investigate their effectiveness. Note that all ablation experiments were performed on SciLitLLM-7B due to budget constraints.

| Benchmark | SciAssess | SciRIFF |
|---|---|---|
| **Model** | | |
| Qwen2.5-Instruct | 46.5 | 50.3 |
| Qwen2.5-Base+SFT | 49.8 | 57.0 |
| Qwen2.5-Base+CPT+SFT | **51.7** | **60.6** |

Table 4: Ablation study of the CPT stage. The results shows that the CPT stage is essential to improve scientific literature understanding ability.

| SFT Dataset | SciAssess | SciRIFF |
|---|---|---|
| Infinity-Instruct only | 47.1 | 51.2 |
| + SciRIFF | 46.8 | 56.7 |
| + SciLitIns | 50.2 | 54.8 |
| + SciRIFF + SciLitIns | **51.7** | **60.6** |

Table 5: Ablation study of SFT data recipes. Our synthetic dataset SciLitIns effectively improves performance on both benchmarks.

| Benchmark | SciAssess | SciRIFF |
|---|---|---|
| **SciLitIns** | | |
| w/o filtering | 50.9 | 60.1 |
| w/ filtering | **51.7** | **60.6** |

Table 6: Ablation study of SFT instruction quality filtering. This shows that our proposed filter refines the quality of SciLitIns .

### 3.3.1 SCIENTIFIC KNOWLEDGE INJECTION VIA CPT (Q2)

We investigate the contribution of the CPT stage for SciLitLLM. We compare the three variants: (1) *Qwen2.5-7B-Instruct:* official instruct-model checkpoint; (2) *Qwen2.5-7B-base + SFT:* applying our SFT stage directly to Qwen2.5-7B-base without CPT; (2) *Qwen2.5-7B-base + CPT + SFT:* SciLitLLM-7B-Instruct.

As shown in Table 4, applying SFT alone to the Qwen2.5-7B-Base model yields improvements on SciAssess (+3.3%) and SciRIFF (+6.7%). This demonstrates that task-specific instruction fine-tuning effectively enhances performance by adapting the model to the specialized tasks in scientific literature understanding. Further, incorporating CPT results in an extra 1.9% improvement on Sci-Assess and a 3.6% gain on SciRIFF. These results highlight the unique role of CPT in pre-adapting the base model to scientific contexts. By curating a high-quality scientific corpus and using it for domain-specific pretraining, the CPT stage injects foundational scientific knowledge into the model. This foundational understanding reduces the gap between general-purpose pretraining and the specific requirements of downstream tasks, allowing the subsequent SFT stage to focus on fine-grained task adaptation rather than compensating for domain knowledge deficiencies.

### 3.3.2 SFT WITH SCILITINS (Q3)

We explore how fine-tuning with our synthetic instruction set, SciLitIns, impacts performance in scientific literature understanding. We incrementally add three datasets to the SFT training set: Infinity-Instruct, SciRIFF, and SciLitIns. As shown in Table 5, using only the general purpose instruction set, Infinity-Instruct, results in the lowest performance on both SciAssess (47.1%) and SciRIFF (51.2%). This result highlights the limitations of general-purpose instructions in addressing the nuanced demands of scientific literature tasks. General-purpose datasets often lack the domain-specific structure, reasoning, and terminology required for tasks like scientific fact verification or molecule generation. These findings emphasize the need for specialized instructions to bridge the gap between general capabilities and domain-specific expertise.

Adding the training set of SciRIFF to Infinity-Instruct improves performance on SciRIFF significantly (+5.5%) but decreases performance on SciAssess (-0.3%). This drop may be due to SciRIFF's relatively narrower focus, primarily covering biomedicine and clinical medicine, while SciAssess spans broader fields such as biology, chemistry, and material science. This highlights the need for a diverse instruction set that covers diverse scientific domains. Replacing SciRIFF with SciLitIns in the fine-tuning process yields better performance than Infinity-Instruct on both benchmarks, achieving 50.2% on SciAssess and 54.8% on SciRIFF. The diverse task coverage in SciLitIns plays a key role in this improvement. By incorporating synthetic instructions from underrepresented domains, SciLitIns broadens the model's exposure to scientific knowledge, enabling it to generalize across varied scientific disciplines.

Finally, combining SciRIFF and SciLitIns yields the best results, with SciAssess reaching 51.7% and SciRIFF achieving 60.6%. These gains suggest that SciLitIns and SciRIFF complement each other, addressing the limitations of each dataset individually. While SciRIFF focuses on highly specific domains like biomedicine, SciLitIns fills the gaps by covering tasks from broader and less-represented scientific areas.

### 3.3.3 IMPACT OF INSTRUCTION FILTER

We conduct an ablation study to assess the impact of quality filter for synthetic instructions by varying whether the dataset SciLitIns was filtered. As discussed in Section 2.2.2, this filter removes low-quality instructions evaluated from five key aspects. Table 6 shows that applying the filter improves the performance of SciLitLLM-7B on SciAssess (+0.8%) and SciRIFF (+0.5%).

While the performance improvements from filtering are evident, the gains are relatively modest. This suggests that the dataset was already of reasonably high quality before filtering, with only a small proportion of low-quality instructions being removed. It also indicates that the fine-tuning process is robust enough to tolerate some noise, although further improvements can be achieved with higher-quality inputs.

## 4 LIMITATIONS

Despite the promising results achieved by SciLitLLM, there are several limitations that should be acknowledged as follows:

- *Insufficient data volume.* Compared with existing pre-training datasets (AI@Meta, 2024; Yang et al., 2024; Taylor et al., 2022), the amount of data used for CPT is comparatively small. While the dataset has been carefully curated to ensure quality and domain relevance, the limited size poses challenges in achieving comprehensive representation across diverse scientific disciplines. With a smaller corpus, certain specialized may be underrepresented, limiting the model's ability to perform well on tasks within these areas. While increasing dataset size is a logical step, it is crucial to maintain high-quality filter to avoid introducing noise. The effectiveness of the current CPT stage demonstrates that a smaller but well-curated dataset can still deliver significant gains. Therefore, the challenge lies in scaling the corpus size while preserving its relevance and quality.

- *Lack of reasoning enhancement.* The current pipeline does not explore advanced reasoning techniques such as Chain-of-Thought (Wei et al., 2022) or Tree-of-Thought (Yao et al., 2023) in the data construction or model inference stages.

- *Lack of preference alignment.* Due to a limited financial budget, the model lacks Reinforcement Learning from Human Feedback (RLHF) (Ouyang et al., 2022). RLHF has shown significant improvements in aligning models with human preferences and ensuring more reliable outputs. Implementing RLHF in future iterations could further enhance the model's reliability.

Addressing these limitations in future research will be crucial developing a more robust and capable LLM specialized in scientific literature understanding.

## 5 CONCLUSION AND FUTURE WORKS

In this paper, we introduce **SciLitLLM**, a specialized model for scientific literature understanding. It is initialized with a general base model – Qwen2.5, and trained through continual pre-training (CPT) and supervised fine-tuning (SFT). For effective scientific knowledge injection during CPT, we propose model-based format and grammar correction method, along with text quality filtering measures. To ensure high-quality and diverse instructions during SFT, we devise instruction synthesis and quality control approaches. Our experiments on widely-used benchmarks demonstrate the effectiveness of this pipeline in adapting a general model to the field of scientific literature understanding. Specifically, SciLitLLM-7B achieves a 4.0% improvement on the SciAssess (Cai et al., 2024) and a 10.1% improvement on the SciRIFF (Wadden et al., 2024) compared to leading models with fewer than 10 billion parameters. SciLitLLM-14B also surpasses leading open-source LLMs with around 70B parameters. We note that this pipeline could be easily adapted to other specialized domains, particularly those lacking adequate high-quality corpora and instruction sets.

Our future work will focus on expanding the diversity and quality of the training data, as well as exploring more efficient methods for domain-specific knowledge injection and high-quality instruction generation. Moreover, we plan to expand our pipeline to include the RLHF (Ouyang et al., 2022) stage for better human preference alignment and safety.

ETHICS STATEMENT

In developing SciLitLLM, we prioritized ethical considerations to ensure the responsible use of our models and methodologies. First, our research does not involve human subjects, and all data used for continual pre-training (CPT) are copyright-compliant. We employed a rigorous quality filtering process to minimize the risk of incorporating biased or misleading content. Nevertheless, we acknowledge that biases inherent in scientific literature, including historical underrepresentation of certain research domains, may propagate into the model's outputs. We also adhere to all applicable legal and ethical research guidelines, such as respecting copyright policies during dataset construction and providing comprehensive model documentation. Our work is conducted with a commitment to research integrity, ensuring that our contributions remain beneficial to the scientific community while addressing ethical responsibilities associated with developing AI technologies.

REPRODUCIBILITY STATEMENT

We have made extensive efforts to ensure the reproducibility of our work. Our proposed SciLitLLM models are available. The corresponding training pipeline have been described in Sections 2. The source code for all modules shown in Figure 5 is included in the supplementary material. The training datasets for both CPT and SFT stages are shown in Table 1.

ACKNOWLEDGEMENT

The work of Sihang Li, Yaorui Shi, and Xiang Wang, is in part supported by the National Natural Science Foundation of China (92270114).

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

## A    RELATED WORKS

### A.1    KNOWLEDGE INJECTION VIA CONTINUAL PRE-TRAINING

Pre-training a language model is usually conducted on a large corpus of textual data to learn the statistical properties of language (Radford et al., 2019; Brown et al., 2020; Lo et al., 2020; Soldaini et al., 2024). To further inject domain knowledge into a general LLM after pre-training, researchers engage in continual pre-training (CPT) on high-quality domain-specific corpora (Jin et al., 2022; Sun et al., 2020), sometimes combined with general corpora. This process enhances the model's fundamental understanding abilities in specific downstream domains while mitigating catastrophic forgetting of general knowledge (Wu et al., 2022; Ke et al., 2023; Mehta et al., 2023). See the comprehensive study (Gupta et al., 2023) for different warm-up strategies for CPT. Additionally, the CPT corpora can be augmented by transforming them into an instruction-response format (Cheng et al., 2023; 2024). Furthermore, the scaling law (Hoffmann et al., 2022) of domain-specific CPT (Que et al., 2024) is explored to determine the optimal mix of data. However, these studies primarily focus on training dynamics and data recipes, leaving the pre-processing for scientific data, especially raw PDF files, largely unexplored. Exhibiting such steps is essential for generating high-quality domain corpora and effectively injecting domain knowledge, representing a significant challenge for practitioners.

### A.2    DOMAIN ADAPTATION VIA SUPERVISED FINE-TUNING

Supervised fine-tuning (SFT) modifies a pre-trained language model to follow specific instructions or perform designated tasks by fine-tuning it on a targeted, task-specific dataset (Raffel et al., 2020; Wei et al., 2021; Liu et al., 2023; Li et al., 2024; Liu et al., 2024; Liao et al., 2024; 2025). Applying SFT to a general LLM for specific domain adaptation has demonstrated effectiveness in various fields: in medicine (Clusmann et al., 2023), corpora of medical literature and clinical notes are used; in law (Cui et al., 2024), legal documents and case law are compiled; and in finance (Wu et al., 2023), financial reports and market data are utilized. In the scientific domain, several studies have specialized LLMs for scientific tasks, often necessitating the construction of a substantial domain-specific dataset with SFT. For example, SciGLM (Zhang et al., 2024a) leverages existing LLMs to generate step-by-step reasoning for unlabelled scientific instructions. ChemLLM (Zhang et al., 2024b), a more specified LLM in the chemistry field, collects structured chemical data from a vast selection of online databases and transforms this structured data into a question-answering format for SFT. SciRIFF (Wadden et al., 2024) converts existing literature understanding datasets into natural language input-output pairs suitable for instruction-tuning using pre-defined templates. However, benchmark studies (Feng et al., 2024; Cai et al., 2024) indicate that SFT alone may not provide adequate scientific knowledge to excel in relevant tasks. This suggests the need for a more comprehensive approach that combines domain knowledge infusion with instruction-following enhancements.

### A.3    LLMS FOR SCIENTIFIC LITERATURE UNDERSTANDING

In the scientific domain (Luo et al., 2024; Liu et al., 2025; Shi et al., 2024), existing strategies for developing specialized LLMs mostly fall into two categories: (1) Supervised fine-tuning with scientific instructions. This approach requires a large, high-quality, and diverse set of instructions to cultivate problem-solving abilities for scientific tasks. Representative works (*e.g.,* SciGLM (Zhang et al., 2024a), ChemLLM (Zhang et al., 2024b), and SciRIFF (Wadden et al., 2024)) have been detailed in Section A.2. (2) Pre-training with scientific corpora. This approach involves pre-training on a large corpus of scientific texts to improve performance on downstream scientific tasks. Early attempts, such as SciBert (Beltagy et al., 2019) and KV-PLV (Zeng et al., 2022), are based on BERT (Devlin et al., 2019) and pre-trained on a large corpus of scientific text for downstream scientific task enhancement. More recently, Galactica (Taylor et al., 2022) is pre-trained on a vast corpus of scientific literature, including research papers, scientific articles, and other relevant scientific texts. Despite these advances, two major limitations hinder these models from excelling in scientific literature understanding: (1) lack of scientific knowledge, and (2) inability to follow task instructions. To address these challenges, we propose a combined pipeline of CPT and SFT to devise a specialized LLM for scientific literature understanding. It injects domain-specific knowledge through CPT while enhanc-

ing task-specific instruction-following abilities through SFT, leading to a more capable LLM for scientific literature understanding.

## B    FORMAT & GRAMMAR CORRECTION EXAMPLES

---

**Raw text parsed by PyPDF2**

Highly p e n e t r a t i n g radiation, such as $\gamma$-rays or fast electorns, deposits ener gy
throughout the solid t a r g e t material. Gas production occurs w i t h i n the solid phase and must d i f f u s e to the surface to be observed.
The a p p a r e n t yield of H
2 can depend on the radiolysis
pro c e d u r e or the particle size if some of the gas remains in the solid. Experiments have shown
that the apparent y i e l d of H 2 can vary by a f a c t o r of 3 in the r a d i o l y s i s of polyethylene spheres
of 7 to 2100 cm2/g (about 9 to 0.03 mm) [12]. The e f f e c t s of gas trapping and diffusion are
not understood in the c o n t e x t of waste storage. Extremely h i g h dose rates in the p r o c e s s i n g of certain materials may lead to
bubble formation, which could a l t e r product quality.
The y i e l d of H
2 in the r a d i o l y s i s of polymers w i t h $\gamma$-rays is well known for several
types of p o l y m e r s [2].

---

**Format & Grammar Correction Prompt**

I have extracted the following raw text from a PDF, but the extraction process has introduced many formatting issues such as unnecessary line breaks, extra spaces, and other artifacts that disrupt the text flow. Could you please help me correct these formatting issues and provide a clean, readable version of the text? Respond with the Corrected Version only.
Raw Text:
{RawText}
Start your response with "Here is the corrected version of the text:".

---

**Text after Format & Grammar Correction**

Highly penetrating radiation, such as $\gamma$-rays or fast electrons, deposits energy throughout the solid target material. Gas production occurs within the solid phase and must diffuse to the surface to be observed. The apparent yield of H2 can depend on the radiolysis procedure or the particle size if some of the gas remains in the solid. Experiments have shown that the apparent yield of H2 can vary by a factor of 3 in the radiolysis of polyethylene spheres of 7 to 2100 cm2/g (about 9 to 0.03 mm) [12]. The effects of gas trapping and diffusion are not understood in the context of waste storage. Extremely high dose rates in the processing of certain materials may lead to bubble formation, which could alter product quality.
The yield of H2 in the radiolysis of polymers with $\gamma$-rays is well known for several types of polymers [2].

---

## C    CPT QUALITY FILTER

We randomly select 50k samples from our CPT data. These selected samples are then scored using the Llama3-70B-Instruct model. The prompt utilized for this scoring process is as follows:

| Filtering Threshold | MMLU-Pro-Bio | MMLU-Pro-Chem | MMLU-Pro-Heal | MaScQA | Avg. |
|---|---|---|---|---|---|
| 15% | 70.59 | 50.32 | 53.14 | 53.14 | 58.32 |
| 25% | 70.45 | 51.21 | 54.06 | 54.06 | 58.75 |
| 35% | 69.82 | 49.52 | 51.64 | 61.64 | 57.67 |

Table 7: 7B base model performance (accuracy %) with different CPT data filtering thresholds.

**Prompt for CPT Data Quality Labelling**

Below is an extract from a textbook. Evaluate whether the text has a high educational value and could be useful in an educational setting for teaching from primary school to grade school levels using the additive 5-point scoring system described below. Points are accumulated based on the satisfaction of each criterion:

- Add 1 point if the extract provides some basic information relevant to educational topics, even if it includes some irrelevant or non-academic content like advertisements and promotional material.
- Add another point if the extract addresses certain elements pertinent to education but does not align closely with educational standards. It might mix educational content with non-educational material, offering a superficial overview of potentially useful topics, or presenting information in a disorganized manner and incoherent writing style.
- Award a third point if the extract is appropriate for educational use and introduces key concepts relevant to school curricula. It is coherent though it may not be comprehensive or could include some extraneous information. It may resemble an introductory section of a textbook or a basic tutorial that is suitable for learning but has notable limitations like treating concepts that are too complex for grade school students.
- Grant a fourth point if the extract is highly relevant and beneficial for educational purposes for a level not higher than grade school, exhibiting a clear and consistent writing style. It could be similar to a chapter from a textbook or a tutorial, offering substantial educational content, including exercises and solutions, with minimal irrelevant information, and the concepts aren't too advanced for grade school students. The content is coherent, focused, and valuable for structured learning.
- Bestow a fifth point if the extract is outstanding in its educational value, and perfectly suited for teaching either at primary school or grade school. It follows detailed reasoning, the writing style is easy to follow, and offers profound and thorough insights into the subject matter, devoid of any non-educational or complex content.

After examining the extract:
- Briefly justify your total score, up to 100 words.
- Conclude with the score using the format: "Educational score: ¡total points¿"

We train a Scientific Texts Quality Classifier on these labeled data samples. The classifier is a 109M BERT (Devlin et al., 2019) classifier, fine-tuned from the checkpoint of fineweb-edu-classifier (Anton et al., 2024). The model is trained for 20 epochs with a learning rate of 0.001 and a batch size of 1024. Ninety percent of the 50K samples are used as the training set, and the rest 10% are used as the validation set. The training process costs approximately 50 minutes on 4 A100 GPUs. We select the checkpoint from the epoch that yields the highest validation micro F1 score as our final checkpoint. During Inference, we set batch size to 2048, and beam number to 1. The inference process costs 90 minutes on 4 A100 GPUs. We utilize the generated to filter out 25% data with the lowest quality.

We also conduct empirical experiments to investigate the sensitivity of model performance to the quality filtering threshold. The results in Table 7 show that setting the filtering threshold at 25% achieves the best overall performance, effectively injecting scientific domain knowledge into the model. When the threshold is set too high (e.g., 35%), the average quality of the corpus improves; however, the significant reduction in data volume limits the ability to fully capture domain knowledge. Conversely, when the threshold is set too low (e.g., 15%), the increase in data volume includes more low-quality content, which negatively impacts model performance. Thus, the trade-off between data quality and quantity is crucial.

# D SFT DETAILS

## D.1 INSTRUCTION GENERATION PIPELINE

In SciLitIns, we focus on generating instructions for three less-represented domains (materials science, medicine, and drug discovery) and five question types:

- **Table Extraction:** Table Extraction tasks evaluate a model's proficiency in extracting, summarizing, and structuring data from an article into a table format.
- **Entity Extraction:** Entity Extraction tasks are designed to evaluate a model's ability to extract specific information, such as entities or relationships, from the text.
- **Molecule Translation:** Molecule Translation tasks evaluate a model's ability to translate molecules between different SMILES formats.

- **Molecule Extraction:** Molecule Extraction tasks ask a model to extract an appropriate molecule from a scientific paragraph that contains multiple molecules.

- **Multiple Choice and True-or-False:** Multiple Choice and True-or-False questions assess a model's ability to select the correct answer from a set of options, testing its knowledge and reasoning on both simple and complex scenarios.

For each of the three scientific domains, we collect a set of high-impact research papers and construct a word frequency table. To generate a question in a given domain, we sample 20 keywords from the corresponding word table and insert them into the prompt for that question. To ensure fair representations of less frequent keywords, we use random sampling with a temperature setting of 3. We will release our code, prompt templates, and word frequency tables. Below is an example of generating a table extraction question:

---

**Prompt for Generating a Table Extraction Question**

I need synthetic training data for training a machine learning model that extracts tables from text correctly. The data should be formatted in JSON, with each entry containing "text" and "answer" attributes. You should generate a paragraph that includes the keywords:
{{keywords}}.
The "text" part must contain enough information for the table to be extracted! In "text" part, You must you include a table description in latex format.
Special notice for the table content:
You should generate a table that has complicated numbers and characters, include non-standard characters, and have a variety of values. Make sure the value you generated do not follow simple patterns, for example, never include deplicate values or values with constant interval in columns.
Your answer should contain as much details as possible. You should only generate one JSON. The value for the two attributes should be two string. Use {{ and }} to warp your output. Pay attention to the escape characters in the latex format. Remember to put a comma at the end of the first string. Never use a json block to wrap your output. Here is the format for your output:
{
"text": "Your paragraph here, remember to include a table in latex format",
"answer": "Your answer table here"
}
Now start your answer:

---

## D.2 INSTRUCTION DEDUPLICATION

The generated synthetic data may contain similar questions or identical answers. To eliminate redundancy, we implement a fuzzy deduplication process using the Levenshtein distance to calculate the similarity score between question-answer pairs. Specifically, for two pairs $(q_1, a_1)$ and $(q_2, a_2)$, their textual similarity is defined as $(1 - \text{lev}(q_1, q_2))(1 - \text{lev}(a_1, a_2))$, where $\text{lev}(\cdot, \cdot)$ denotes the Levenshtein distance. Due to significant differences between texts from different question types, we compute similarity matrices separately for each type. We then use a disjoint-set data structure to merge highly similar data points. We use this process to remove approximately 5% to 10% of duplicated data for each question type.

## D.3 QUALITY ASSESSMENT OF GENERATED SFT INSTRUCTIONS

In section 2.2.2, we sample 10k instruction pairs from SciLitIns and evaluate them by Llama-3-70B using the below prompt. Specifically, the quality is evaluated from five aspects: clarity, complexity, correctness, usefulness, and adaptability, assigning each instruction a score from 0 (lowest quality) to 5 (highest quality).

---

**SFT Evaluation Prompt**

You are a helpful and precise assistant for checking the quality of instruction-tuning data for large language models. Your task is to evaluate the given instruction using the criterions described below.
- Clarity: The sample should be clear, specific, and unambiguous, providing a well-defined task for the model to perform.
- Complexity: The sample should be advanced complexity that necessitate a high level of comprehension and cognitive processing, challenging the language model significantly.
- Correctness: The sample is impeccably written, with flawless grammar, syntax, and structure, demonstrating exceptional clarity and professionalism.
- Usefulness: The sample should be highly useful, and contribute to expanding the model's knowledge base.
- Adaptability: The sample could be adapted to different contexts or use cases, showing some flexibility.
After examining the instruction-response pair:
- Briefly justify your scores with a paragraph in the field "Explanation", up to 500 words.
- For each point of criterion above, assign a score from 1 to 5.
- You should only provide the rest of your answer in a structured format as shown below, and make sure your response can be directly parsed by computer programs.
Below is a template for your response:
Explanation: ¡string, your explanations to the scores¿
====================
{
"Clarity": <int, complexity_score>,
"Complexity": <int, complexity_score>,
"Correctness": <int, quality_score>,
"Usefulness": <int, usefulness_score>,
"Adaptability": <int, adaptability_score>,
"Total": <int, total_score>
}

---

Below is an example of SciLitIns, which will be sent to Llama-3-70B for evaluation.

---

**An Example in SciLitIns**

**Context**

In recent studies regarding material science, the crmatrix has shown significant importance in understanding fracture behaviors. Alternatively, many researchers have proposed a series of experiments to analyze these phenomena. Even though the process is delicate, the enthusiasm for discovering new findings is understandable. The following Table 1 presents a schematic illustration of the enthalpy changes observed during the different fracture phases. These studies are indispensable for advancing our knowledge of the material's resilience.

In addition, in certain cases, pronounced errors in measurement can occur, but the references provided can guide through the correct methodology. Prime among the tools used is the differential scanning calorimetry which helps measure these enthalpy changes accurately.

```
\begin{table}[h!]\n\centering\n\begin{tabular} {|c|c|c|c|}\n\hline\nMaterial & Initial
    Enthalpy (kJ/mol) & Final Enthalpy (kJ/mol) & Fracture Phase \\\n\hline\nMaterial A &
    25.4 & 47.8 & Brittle \\\n\hline\nMaterial B & 22.3 & 45.0 & Ductile
    \\\n\hline\nMaterial C & 28.9 & 50.2 & Semi-brittle
    \\\n\hline\n\end{tabular}\n\caption{Enthalpy changes during fracture
    phases}\n\label{table:enthalpy}\n\end{table}
```

The process of accurately measuring these values is crucial for the material selection in various engineering applications, making it an indispensable research area.

**Instruction**
Extract the table in the paragraph.

**Response**
Material,Initial Enthalpy (kJ/mol),Final Enthalpy (kJ/mol),Fracture Phase
Material A,25.4,47.8,Brittle
Material B,22.3,45.0,Ductile
Material C,28.9,50.2,Semi-brittle

---

Below is the evaluation by Llama-3-70B on the example from SciLitIns. Each criterion is scored out of 5 points, making the total possible points 25. The total score is then averaged, resulting in an overall score out of 5.

---

**Evaluation by Llama-3-70B**

Explanation:
The instruction is clear and specific, asking the model to extract a table from a given paragraph. The paragraph itself is well-written, with proper grammar and syntax, making it easy to understand. The table is also well-structured and provides valuable information about enthalpy changes during fracture phases. The task requires a good understanding of the text and the ability to identify the table. However, the complexity of the task is relatively low, as it only involves extracting a table, and the context is not particularly nuanced or ambiguous. The task is useful for advancing knowledge in material science, and the table could be adapted to different contexts or use cases.
====================
"Clarity": 5,
"Complexity": 2,
"Correctness": 5,
"Usefulness": 4,
"Adaptability": 4,
"Total": 20

---

# E    BENCHMARK DETAILS

To the best of our knowledge, there are two commonly-adopted datasets for scientific literature understanding:

- SciRIFF (Wadden et al., 2024) evaluates essential scientific literature understanding capabilities, including information extraction, summarization, question answering, claim verification, and classification. Data points in SciRIFF are notable for their long input contexts and complicated structured outputs. The Qasper and SciFact tasks have two different evaluation methods and thus two results. For MuP and one evaluation method of Qasper, we use GPT-4o as language model evaluator, while the original paper uses GPT-3.5. We note that SciRIFF contains a separate training set used in the SFT stage in our study.

- SciAssess (Cai et al., 2024) features an end-to-end benchmark of understanding PDF content. It includes 28 tasks from four scientific domains: biology, chemistry, materials and medicine. SciAssess is used exclusively for testing in our evaluation.

Overall, SciRIFF provides basic benchmarks for comprehending short scientific segments and various instructions, while SciAssess presents more challenging tasks involving longer contexts from raw PDFs.

# F    DETAILED PERFORMANCE ON SCIASSESS

The detailed results on each task in SciAssess are shown in Table 8.

| Domain | Task | SciTulu-7B | Mistral-7B | Llama3.1-8B | Qwen2.5-7B | SciLitLLM-7B | Qwen2.5-14B | Llama3-70B | Qwen2.5-72B | SciLitLLM-14B | GPT3.5 | GPT4o |
|---|---|---|---|---|---|---|---|---|---|---|---|---|
| Biology | Average | 45.3 | 52.0 | 63.4 | 60.9 | 65.3 | 65.9 | 69.4 | 67.2 | 67.0 | 55.4 | 68.9 |
|  | MmluPro | 41.4 | 34.4 | 68.9 | 70.2 | 72.2 | 77.7 | 81.5 | 84.0 | 79.6 | 65.0 | 87.4 |
|  | ChartQA | 35.2 | 40.7 | 46.7 | 47.7 | 48.2 | 51.8 | 47.7 | 48.7 | 56.3 | 31.2 | 55.8 |
|  | ChemRecog | 54.9 | 56.0 | 73.7 | 66.7 | 84.3 | 80.1 | 83.6 | 76.4 | 77.8 | 64.9 | 79.5 |
|  | CompRecog | 41.6 | 72.4 | 69.3 | 66.8 | 72.2 | 74.5 | 76.8 | 71.2 | 71.1 | 63.6 | 73.3 |
|  | DiseRecog | 58.0 | 67.6 | 76.2 | 57.4 | 76.1 | 64.7 | 79.3 | 79.3 | 74.1 | 68.8 | 76.3 |
|  | GeneFunc | 40.6 | 40.7 | 45.8 | 56.6 | 38.9 | 46.4 | 47.4 | 43.8 | 43.2 | 39.1 | 41.0 |
| Chemistry | Average | 19.0 | 34.0 | 46.1 | 47.9 | 55.4 | 60.0 | 62.5 | 63.7 | 63.4 | 37.3 | 68.0 |
|  | MmluPro | 13.8 | 17.9 | 40.8 | 46.2 | 52.6 | 63.7 | 67.6 | 72.3 | 63.2 | 30.3 | 74.5 |
|  | ElecQA | 19.6 | 41.7 | 53.8 | 63.3 | 70.4 | 75.9 | 75.5 | 78.5 | 78.4 | 30.5 | 85.5 |
|  | OledExtr | 0.9 | 22.7 | 13.7 | 36.3 | 30.6 | 42.0 | 56.3 | 49.9 | 37.4 | 28.0 | 43.8 |
|  | ChartQA | 33.3 | 46.7 | 86.7 | 46.7 | 66.7 | 80.0 | 80.0 | 80.0 | 80.0 | 66.7 | 86.7 |
|  | PolyQA | 25.0 | 56.7 | 59.1 | 76.9 | 88.0 | 90.9 | 85.2 | 91.4 | 91.3 | 33.0 | 93.8 |
|  | PolyExtr | 7.4 | 35.8 | 43.1 | 42.6 | 66.4 | 52.9 | 69.0 | 69.2 | 78.1 | 56.2 | 75.9 |
|  | SolExtr | 31.5 | 33.7 | 33.2 | 40.5 | 36.1 | 41.1 | 44.7 | 43.7 | 41.3 | 40.8 | 44.4 |
|  | ReactQA | 21.0 | 23.1 | 30.3 | 32.8 | 33.3 | 39.0 | 38.5 | 37.9 | 51.3 | 27.2 | 48.7 |
|  | MechQA | 18.2 | 27.3 | 54.5 | 45.5 | 54.5 | 54.5 | 45.5 | 50.0 | 50.0 | 22.7 | 59.1 |
| Material | Average | 31.3 | 36.4 | 50.9 | 48.9 | 53.7 | 57.9 | 59.2 | 59.4 | 62.6 | 37.0 | 62.0 |
|  | MatQA | 36.9 | 43.0 | 59.7 | 56.7 | 58.6 | 66.5 | 73.8 | 71.9 | 70.3 | 52.1 | 76.8 |
|  | ChartQA | 33.3 | 66.7 | 66.7 | 73.3 | 46.7 | 60.0 | 46.7 | 53.3 | 53.3 | 40.0 | 46.7 |
|  | CompExtr | 8.0 | 8.9 | 9.9 | 13.9 | 40.6 | 37.1 | 45.7 | 43.0 | 42.9 | 18.9 | 46.2 |
|  | TempQA | 24.2 | 32.9 | 68.1 | 43.0 | 72.0 | 69.6 | 65.2 | 64.7 | 79.2 | 29.5 | 80.7 |
|  | SampDiff | 29.1 | 20.7 | 36.3 | 39.7 | 44.3 | 52.7 | 62.4 | 57.8 | 60.8 | 32.9 | 62.4 |
|  | TreatSeq | 56.4 | 46.5 | 64.9 | 66.8 | 59.9 | 61.4 | 61.4 | 65.8 | 68.8 | 48.5 | 59.4 |
| Medicine | Average | 19.9 | 25.1 | 30.2 | 28.3 | 32.4 | 31.5 | 37.4 | 37.8 | 39.9 | 31.8 | 45.8 |
|  | MmluPro | 21.3 | 30.3 | 58.4 | 51.7 | 57.2 | 62.0 | 71.0 | 68.5 | 65.8 | 53.1 | 76.3 |
|  | AffiExtr | 2.7 | 3.7 | 3.7 | 2.6 | 5.6 | 4.0 | 4.7 | 7.1 | 3.9 | 5.5 | 10.1 |
|  | ChartQA | 33.3 | 40.0 | 26.7 | 40.0 | 33.3 | 33.3 | 40.0 | 33.3 | 53.3 | 33.3 | 46.7 |
|  | TagMol | 1.2 | 0.2 | 7.1 | 0.7 |  | 6.2 | 14.3 | 13.6 | 7.3 | 2.3 | 22.9 |
|  | MarkMol | 6.9 | 20.2 | 35.2 | 20.8 | 37.5 | 23.6 | 42.5 | 44.3 | 47.1 | 52.3 | 58.5 |
|  | MolDoc | 54.0 | 56.0 | 50.0 | 54.0 | 58.0 | 60.0 | 52.0 | 60.0 | 62.0 | 44.0 | 60.0 |

Table 8: Detailed model performance on SciAssess tasks.

| Eval Datasets | In-house Textbooks (%) | In-house Journals (%) |
|---|---|---|
| SciRIFF | 1.1 | 0.7 |
| SciAssess | 11.0 | 1.1 |

Table 9: Contamination rates in textbook and journal datasets.

| Category | Contamination Rates (%) | SciLitLLM-7B | | SciLitLLM-14B | |
|---|---|---|---|---|---|
| | | Full Set (%) | Clean Set (%) | Full Set (%) | Clean Set (%) |
| Biology | 12.2 | 65.9 | 66.3 | 67.5 | 67.6 |
| Chemistry | 13.3 | 54.4 | 55.8 | 64.2 | 65.2 |
| Material | 12.5 | 52.5 | 50.7 | 61.8 | 62.9 |
| Medicine | 8.7 | 33.6 | 34.2 | 42.4 | 43.9 |
| Mean | 12.0 | 51.6 | 51.6 | 59.0 | 59.8 |

Table 10: Performance comparison of SciLitLLM on full and clean evaluation sets of SciAssess.

## G  CONTAMINATION STUDY

To investigate data contamination in our datasets and its potential impact on downstream performance, we conducted a contamination analysis in this section. Prior studies consistently indicate that data contamination in pre-training datasets has negligible effects on evaluation performance:

- The GPT-4 technical report states: *Contamination overall has very little effect on the reported results* (OpenAI, 2023). For instance, tasks such as AP US History (73% contamination rate), AP World History (47% contamination rate), and LSAT (39% contamination rate) show negligible differences in performance between the full evaluation set and a clean evaluation set with all contaminated data points removed.

- PaLM analyzed datasets with contamination rates ranging from 20% to 75% and concluded that *data contamination does not cause meaningful inflation of our reported results* (Chowdhery et al., 2023).

For our contamination analysis, we adopt the contamination detection method described in the GPT-4 technical report: For each evaluation instance, we randomly extract three substrings of 50 characters each. A match is determined if any of the three sampled substrings appears as a part of the processed training example.

The contamination rates for our textbook and journal datasets are summarized in Table 9. Our analysis reveals that contamination rates are very low for SciRIFF but higher for SciAssess, particularly in the textbook dataset. To evaluate whether contamination impacts model performance, we assessed SciLitLLM on SciAssess using both the full evaluation set and a clean evaluation set (with all contaminated examples removed). As shown in Table 10, the differences in performance between the full and clean evaluation sets are minimal, consistent with prior findings. These results reinforce the conclusion that data contamination does not significantly affect the performance of SciLitLLM.

## H  HUMAN EVALUATION OF DATASET QUALITY

To further validate the effectiveness of the proposed framework for creating the CPT and SFT datasets, we conducted a human evaluation study. This study involved sampling random subsets of unfiltered CPT and SFT data and comparing human annotations to the scores generated by our quality filters.

For the CPT dataset, we sampled 50 entries from the textbook and journal datasets. For the SFT dataset, we sampled 50 entries from SciLitIns. Each sampled entry was independently evaluated by four annotators using the same prompts provided to the LLMs. To assess the alignment between human annotations and the quality filter, we computed two measures:

- Human-Human Agreement: Calculated as the average Spearman correlation coefficient across all pairs of annotators.
- Human-Quality Filter Agreement: Calculated as the average Spearman correlation between each annotator's scores and the quality filter's score.

A higher Spearman correlation indicates a stronger agreement between the compared sets of scores. The evaluation results are summarized in Table 11. The results show that for both datasets, the Human-Quality Filter Agreement is comparable to, or even exceeds, the Human-Human Agreement. This indicates that the quality filter closely aligns with human annotations, demonstrating its reliability in capturing quality as perceived by humans.

| Agreement | Human-Human | Human-Quality Filter |
|-----------|-------------|----------------------|
| CPT       | 0.58        | 0.76                 |
| SFT       | 0.89        | 0.88                 |

Table 11: Comparison of Human-Human Agreement and Human-Quality Filter Agreement for CPT and SFT datasets.

