# OpenReview forum: "SciLitLLM: How to Adapt LLMs for Scientific Literature Understanding"
_ICLR.cc/2025/Conference — ICLR 2025 Poster_

### Official Review · Reviewer_fcV1 · 2024-11-03

**Soundness:** 3
**Presentation:** 3
**Contribution:** 3
**Rating:** 8
**Confidence:** 3

**Summary:**

The paper introduces SciLitLLM, a specialized language model for scientific literature understanding, built using a hybrid approach combining continual pre-training (CPT) and supervised fine-tuning (SFT). The key contributions include 1) A pipeline that combines CPT with high-quality scientific corpora and SFT with diverse scientific instructions, 2) Novel methods for improving scientific text quality and generating domain-specific instructions, and 3)  Empirical results showing improved performance on scientific understanding benchmarks SciRIFF and SciAssess

**Strengths:**

### Well-Motivated Approach
- The hybrid CPT+SFT strategy effectively addresses both domain knowledge and task-specific capabilities.
- The pipeline is well-designed with clear motivation for each component.
- The approach is generalizable to other specialized domains

### Technical Contributions
>Pipeline includes innovative components like LLM-based format correction (Section 3.1.1) and quality filtering (Section 3.1.2)
The instruction synthesis method (Section 3.2.1) is clever and tackles the challenge of limited scientific instruction data.

### Solid empirical results
- SciLitLLM-7B outperforms similar-sized models by significant margins \~4% on SciAssess, +\~10% on SciRIFF.
- SciLitLLM-14B surpasses larger proprietary instruction tuned models (70B+ parameters).

**Weaknesses:**

### Weaknesses
The CPT corpus (12.7B tokens) is relatively small compared to standard pre-training datasets The paper acknowledges this limitation but could discuss potential impacts more thoroughly. For example, how this affect the representation from different scientific subject domains.

Otherwise I don't see clear weakness of paper of such kind. The paper appears comprehensive and well-executed for the research scope.

**Questions:**

1. How sensitive is the model performance to the quality filtering threshold (currently set at 25%)? Was this choice empirically validated?

2. The instruction synthesis method uses GPT-4 - have you explored using smaller models or your own models in a bootstrapping approach?

---

> ### Author Response · Authors · 2024-11-21
>
> > Q1: The CPT corpus (12.7B tokens) is relatively small compared to standard pre-training datasets. The paper acknowledges this limitation but could discuss potential impacts more thoroughly. For example, how this affects the representation from different scientific subject domains.
>
> R1: Thank you for your insightful feedback! We agree that discussing the implications of the relatively small scale of our CPT corpus is crucial for a comprehensive understanding of the work’s limitations.
>
> Specifically, while our dataset has been meticulously curated to ensure high quality and domain relevance, the limited size presents challenges in achieving comprehensive representation across diverse scientific disciplines. For example, certain specialized domains may be underrepresented, potentially limiting the model’s performance on tasks requiring expertise in those areas.
>
> While increasing the dataset size is a logical next step, it is equally important to maintain stringent quality control to avoid introducing noise, which could dilute the benefits of domain-specific data. The promising performance observed in our experiments highlights that a smaller but well-curated dataset can still yield significant gains. However, scaling the corpus effectively while preserving relevance and quality remains a key challenge for future work.
>
> We have included this expanded discussion in the revised Limitations section of the paper. We appreciate your thoughtful suggestion and hope these additions address your concerns.
>
> > Q2: How sensitive is the model performance to the quality filtering threshold (currently set at 25%)? Was this choice empirically validated?
>
> R2: Thank you for your constructive feedback. We have conducted empirical experiments to investigate the sensitivity of model performance to the quality filtering threshold. The results are summarized below:
>
> | Filtering Threshold | MMLU-Pro-Bio | MMLU-Pro-Chem | MMLU-Pro-Heal | MaScQA | Avg.   |
> |---------------------|--------------|---------------|---------------|--------|--------|
> | 15%                | 70.59        | 50.32         | 53.14         | 59.24  | 58.32  |
> | 25%                | 70.45        | 51.21         | 54.06         | 59.27  | 58.75  |
> | 35%                | 69.82        | 49.52         | 51.64         | 59.71  | 57.67  |
>
> The results show that setting the filtering threshold at 25% achieves the best overall performance, effectively injecting scientific domain knowledge into the model.
>
> - When the threshold is set too high (e.g., 35%), the average quality of the corpus improves; however, the reduction in data volume limits the ability to fully capture domain knowledge.
> - Conversely, when the threshold is set too low (e.g., 15%), the increase in data volume includes more low-quality content, which negatively impacts model performance.
>
> As you noted, the trade-off between data quality and quantity is crucial. We have included this analysis in Appendix C of the revised paper to clarify our choice of the 25% threshold. Thank you for highlighting this important aspect!
>
> > Q3: The instruction synthesis method uses GPT-4 - have you explored using smaller models or your own models in a bootstrapping approach?
>
> R3: Thank you for your insightful comment! We agree that bootstrapping strategies, such as leveraging smaller models or our own models for data synthesis, could be a promising direction for future improvements. These approaches effectively address data scarcity by iteratively enlarging the dataset, as demonstrated in influential works like [1][2].
> We also recognize the critical role of high-quality data in scientific domains, and we are excited to explore this direction in the next version of SciLitLLM. However, given our current computational constraints and the need to prioritize other experiments during this rebuttal period, we may not be able to implement and evaluate this approach in the current submission cycle.
>
> We appreciate your understanding and believe this suggestion will be valuable for guiding the continued development of our model. Thank you for highlighting this important area for improvement!
>
> [1] BLIP: Bootstrapping Language-Image Pre-training for Unified Vision-Language Understanding and Generation
>
> [2] BLIP-2: Bootstrapping Language-Image Pre-training with Frozen Image Encoders and Large Language Models

---

> > ### Comment · Reviewer_fcV1 · 2024-12-01
> > **Response to authors**
> >
> > I thank the authors for clarifying my questions. My evaluation well reflect my acknowledgement of this work.

---

> > > ### Author Response · Authors · 2024-12-02
> > > **Thank you for increasing the encouraging feedbacks**
> > >
> > > Dear Reviewer **fcV1**:
> > >
> > > Your positive feedback means a great deal to us and validates the effort we put into our work and our rebuttal. And your constructive and thoughtful comments have been incredibly helpful, and we are truly grateful for your support.
> > >
> > > Authors

---

### Official Review · Reviewer_W9HG · 2024-11-03

**Soundness:** 3
**Presentation:** 3
**Contribution:** 3
**Rating:** 6
**Confidence:** 4

**Summary:**

The paper proposes a new strategy to improve LLM for scientific literature understanding, which includes continual pretraining and supervised fine-tuning. The paper uses Llama3-8B to correct parsing errors and filters the dataset with Llama3-70B. The paper continues pretraining the Qwen2.5.The paper designs a three-step pipeline to generate diverse scientific contexts and corresponding QA pairs. The paper then incorporates heuristic deduplication and LLM-based filtering for the instructions. The proposed framework seems to improve the performance compared to SciRIFF. The paper also includes an ablation study.

**Strengths:**

1. The paper proposes a new framework that includes continual pretraining and supervised fine-tuning. The proposed framework and dataset can be very useful for other LLMs specialized in scientific understanding. The released dataset seems to be very comprehensive compared to the existing dataset.
2. The paper shows that with the new framework, the paper can further improve the performance of general LLM. The framework is especially useful in small models. Additionally, the paper includes an ablation study for CPT, SFT, and instruction quality filters.
3. The paper provides the code and its model. In the appendix, the paper shows the improvement of format & grammar correction, CPT quality filter, SFT details, benchmark details, and detailed performance on SciAssess.

**Weaknesses:**

1. The creation of the CPT and SFT datasets seems to rely on LLMs. The paper can randomly sample a small subset of the created dataset to check its quality with humans to show the effectiveness of the proposed framework.
2. The experiment and ablation study is not comprehensive. The paper fails to show that the model is fine-tuned to other existing scientific understanding datasets, such as the Dolma dataset (https://allenai.github.io/dolma/). The paper did include SciTulu-7B dataset; however, scitulu is based on LLama2-tb. The paper should also show the model's performance only finetuned with SciLitIns. The analysis in the experiment is also rather simple. The paper needs to provide some explanation instead of just repeating the results in the table.
3. Some details are not very clear. What is the score used in Tables 2 and 3? Many additional evaluation results and analysis are put in the Appendix. Authors should move some of them to the main paper.

**Questions:**

What is the difference between the new CPT dataset and the Dolma Dataset (https://allenai.github.io/dolma/)?

---

> ### Author Response · Authors · 2024-11-21
>
> > Q1: The creation of the CPT and SFT datasets seems to rely on LLMs. The paper can randomly sample a small subset of the created dataset to check its quality with humans to show the effectiveness of the proposed framework.
>
> Thank you for the suggestion to evaluate the quality of the CPT and SFT datasets with human annotations! In response, we sample random subsets of unfiltered CPT and SFT data and ask human annotators to rate their quality. We then compare the human annotations to the scores from our quality filters to validate the effectiveness of the proposed filtering framework.
>
> **Evaluation Setup**. For the CPT dataset, 50 entries are sampled from textbook and journal datasets. For the SFT dataset, 50 entries are sampled from SciLitIns. Each entry is independently evaluated by four annotators using the same prompts provided to LLMs. To assess alignment with human evaluations, we compute two measures:
>
> - Human-Human Agreement: Calculated as the average Spearman correlation coefficient across all pairs of annotators.
> - Human-Quality Filter Agreement: Calculated as the average Spearman correlation between each annotator’s scores and the quality filter's score.
>
> A higher Spearman correlation indicates a stronger agreement between the compared sets of scores. The results are presented below:
>
> | Dataset | Human-Human Agreement | Human-Quality Filter Agreement |
> |---------|------------------------|--------------------------------|
> | CPT     | 0.58                  | 0.76                           |
> | SFT     | 0.89                  | 0.88                           |
>
> On both datasets, the Human-Quality Filter Agreement is comparable to or even exceeds the Human-Human Agreement. This indicates that the scores generated by our quality filter are closely aligned with human annotations, demonstrating the reliability of the filtering framework in capturing quality as perceived by humans.
>
> We hope this answers your question! Detailed results and analyses are included in the Appendix H.
>
> > Q2: The experiment and ablation study is not comprehensive. The paper fails to show that the model is fine-tuned to other existing scientific understanding datasets, such as the Dolma dataset (https://allenai.github.io/dolma/).
>
> R2: Thank you for bringing up the Dolma dataset. We appreciate the opportunity to clarify our position and the distinctions between our dataset and existing resources.
>
> The Dolma dataset, designed as a broad pretraining corpus, includes a wide variety of topics, with its scientific subset identified as the peS2o dataset [1]. To provide a clearer comparison, we outline key characteristics of the peS2o dataset and our scientific textbook and journal dataset below:
>
> | Dataset                          | #Tokens | Description                                                                                  |
> |--------------------------------------|---------|----------------------------------------------------------------------------------------------|
> | PeS2o dataset                        | 42.0B   | Open access academic papers (including arXiv, PubMed, etc.)                                  |
> | Our in-house textbook and journal dataset | 13.7B   | Scientific Textbooks and journals with copyrights                                            |
>
> While the peS2o dataset offers broad coverage of open-access academic papers, we would respectfully note that such open-access materials (e.g., books and academic papers) may already be included in the pretraining corpus of Qwen [2], as mentioned in the Qwen technical report: “Our dataset is designed to meet these requirements and includes public web documents, encyclopedia, books, codes, etc.”
>
> In contrast, **our dataset provides unique value through its curated, high-quality content from copyrighted textbooks and journals**, which is specifically tailored to scientific literature tasks. Although we cannot release the dataset directly due to copyright restrictions, we have shared a complete list of the textbooks used, including textbook titles, authors, and other meta-information (See the `books.xlsx` file in the updated supplementary materials).
>
> Thank you again for your constructive feedback! We have also included references to these papers in the related work section.
>
> [1] Luca Soldaini and Kyle Lo. 2023. *peS2o (Pretraining Efficiently on S2ORC) Dataset.* (https://github.com/allenai/peS2o)
>
> [2] *Qwen Technical Report* (https://arxiv.org/abs/2309.16609)

---

> > ### Author Response · Authors · 2024-11-21
> >
> > > Q3: The paper did include SciTulu-7B dataset; however, SciTulu is based on LLaMA-2-tb.
> >
> > R3: Thank you for your insightful comments. We appreciate your concern regarding the fairness of the comparison with SciTulu.
> >
> > We included SciTulu in Table 3, despite the model family difference, as it is currently the only official checkpoint available for scientific literature understanding. We hope this provides clarity and addresses your concerns.
> >
> > We agree that a fair comparison is essential to accurately assess the effectiveness of our SFT data, SciLitIns. SciTulu is based on the LLaMA-2 and is fine-tuned on both general-purpose data (Tulu v2 SFT mix) and domain-specific data (SciRIFF). For a clearer comparison, we also included a variant of SciLitLLM (Table 5, line 2) that uses the Qwen2.5 model fine-tuned on both general-purpose data (Infinity Instruct) and domain-specific data (SciRIFF). SciLitLLM outperforms this variant, which highlights the effectiveness of SciLitIns.
> >
> > > Q4: The paper should also show the model's performance only fine-tuned with SciLitIns.
> >
> > R4: Thank you for your thoughtful comment! We have revised the paper to include the performance results for the model fine-tuned only on SciLitIns, as shown in the updated Table 5 below:
> >
> > | SFT Dataset                   | SciAssess | SciRIFF |
> > |-------------------------------|-----------|---------|
> > | Infinity-Instruct only        | 47.1      | 51.2    |
> > | + SciRIFF                     | 46.8      | 56.7    |
> > | + SciLitIns                   | 50.2      | 54.8    |
> > | + SciRIFF + SciLitIns         | 51.7      | 60.6    |
> >
> > Specifically, fine-tuning on SciLitIns alone yields better performance than using only Infinity-Instruct, achieving 50.2% on SciAssess and 54.8% on SciRIFF. These results highlight the value of SciLitIns as a synthetic instruction set tailored to scientific literature tasks.
> >
> > We hope this addition addresses your feedback and further demonstrates the contribution of SciLitIns. Thank you again for your suggestion!
> >
> > > Q5: The analysis in the experiment is also rather simple. The paper needs to provide some explanation instead of just repeating the results in the table.
> >
> > R5: Thank you for your valuable feedback. We have revised the paper and included your suggested revisions to provide deeper analysis in the Experiment section, beyond merely reporting the results from the tables (Page 8-9). We hope this improved analysis meets your expectations and demonstrates the robustness of our methodology. Thank you for your constructive suggestions!
> >
> > > Q6: Some details are not very clear. What is the score used in Tables 2 and 3?
> >
> > R6: Thank you for your comment! We have revised the paper to address this point. Specifically:
> >
> > - For Table 2, we now explicitly state in the caption that all metrics reported are accuracy (%).
> > - For Table 3, since the tasks in SciRIFF [3] and SciAssess [4] involve diverse metrics such as F1, accuracy, BLEU, and LLM judge scores, we have added references in the benchmark introduction paragraph to guide readers to the original papers for detailed explanations of these metrics.
> >
> > We hope this clarification resolves your concerns, and we appreciate your feedback in helping us improve the paper.
> >
> >
> > > Q7: Many additional evaluation results and analysis are put in the Appendix. Authors should move some of them to the main paper.
> >
> > R7: Thank you for your valuable suggestion. We have revised our paper by moving the evaluation results for both CPT and SFT data quality from the Appendix to the main body. We believe this adjustment will provide readers with a clearer and more comprehensive understanding of the effectiveness of our proposed data processing pipeline.
> >
> > We appreciate your feedback and hope the revised organization enhances the clarity of our work.
> >
> > > Q8: What is the difference between the new CPT dataset and the Dolma dataset?
> >
> > R8: Thank you for the question! Please refer to Q2.
> >
> > [3] SciRIFF: A Resource to Enhance Language Model Instruction-Following over Scientific Literature
> >
> > [4] SciAssess: Benchmarking LLM Proficiency in Scientific Literature Analysis

---

> > > ### Comment · Reviewer_W9HG · 2024-11-21
> > >
> > > Thank you very much for your reply! I raised my score to 6.

---

> > > > ### Author Response · Authors · 2024-11-22
> > > > **Thank you for increasing the rating and the encouraging feedbacks**
> > > >
> > > > Thank you for increasing the rating from 5 to 6. Your positive feedback means a great deal to us and validates the effort we put into our work and our rebuttal.

---

### Official Review · Reviewer_7iRT · 2024-11-03

**Soundness:** 2
**Presentation:** 2
**Contribution:** 2
**Rating:** 5
**Confidence:** 4

**Summary:**

This paper proposes a synthetic data generation pipeline for training
LMs for scientific literature understanding tasks. The main
contributions include:

1.  a pipeline for curating continued pre-training corpus based on
    textbooks and research papers (including document parsing,
    formatting, and filtering),

2.  another pipeline for creating instruction fine-tuning data for
    scientific literature understanding tasks via prompting GPT-4o.

3.  the authors show that finetuning Qwen-2.5 model (7B and 14B) on the
    CPT and SFT data can improve the performance on scientific
    literature understanding tasks.

**Strengths:**

1.  Overall the paper is nicely written and the pipeline is nicely
    presented.

2.  The curated dataset and models could be helpful.

**Weaknesses:**

1.  The primary contribution of the paper seems to be focused on the
    dataset construction; overall the method resembles similar works for
    synthetic data generation and there are some limitations:

    1.  The PDF processing pipeline can be improved. Scientific PDF are
        known to contain complex layout and structures, and previous
        work have identified that using simple PDF parsers can lead to
        suboptimal training results (S2ORC, Lo and Wang, and ). However,
        the author primarily uses a simple PyPDF parser (\"Converting
        these documents using tools like PyPDF2 often introduces
        formatting and syntax errors, which degrade the quality of the
        corpus (line 246)\"). I'd suggest the authors investigate the
        text quality issues and check other libraries like papermage, Lo
        et al.

2.  The results are not very strong. I'd imagine a domain-specifically
    distilled model can have a substance gain in performance compared to
    GPT-4o, especially the instruction fine-tuning dataset is generated
    via GPT-4o (line 331); however, as shown in table 3, the trained
    models (SciLitLLM-7B and SciLitLLM-14B) are on par with GPT-4o. Also
    the experimental design and presentation could be improved (see my
    suggestions in questions).

**Questions:**

1.  Are there special reasons why we should group the results based on
    the 10B model size? (table 2/3). I think it's more reasonable to
    organize based on pre-trained only/with instruction tuning/with
    domain specific tuning?

2.  Also in this paper there is only fine-tuning results on the Qwen
    model family but not others. It would be interesting to compare the
    fine-tuning effects on llama or other model families.

    1.  the author compared the performance with the SciTulu model,
        which is trained based on Llama-2 families. I don't think it's a
        fair comparison in table 3.

---

> ### Author Response · Authors · 2024-11-21
>
> > Q1: The PDF processing pipeline can be improved. Scientific PDF are known to contain complex layout and structures, and previous work have identified that using simple PDF parsers can lead to suboptimal training results (S2ORC, Lo and Wang, and ). However, the author primarily uses a simple PyPDF parser ("Converting these documents using tools like PyPDF2 often introduces formatting and syntax errors, which degrade the quality of the corpus (line 246)"). I'd suggest the authors investigate the text quality issues and check other libraries like papermage, Lo et al.
>
> R1: Thank you for this valuable feedback regarding PDF processing. We compared a few PDF parsing options, focusing primarily on the closed-source Mathpix and the open-source PyPDF2 library. Due to the high costs associated with Mathpix (approximately $80,000 to process 70,000 textbooks), we opted for PyPDF2 along with additional formatting and syntax error correction techniques described in the paper to mitigate common parsing issues.
>
> We appreciate your suggestion to explore alternatives like Papermage [1], and will investigate these in our future work to further improve parsing quality. Given our current computational limitations, however, we may not be able to retrain the model with Papermage-parsed data within the rebuttal period. We hope for your understanding in this regard.
>
> [1] PaperMage: A Unified Toolkit for Processing, Representing, and Manipulating Visually-Rich Scientific
>
> > Q2: The results are not very strong. I'd imagine a domain-specifically distilled model can have a substance gain in performance compared to GPT-4o, especially the instruction fine-tuning dataset is generated via GPT-4o (line 331); however, as shown in table 3, the trained models (SciLitLLM-7B and SciLitLLM-14B) are on par with GPT-4o.
>
> R2: Thank you for your thoughtful feedback. We would like to take this opportunity to clarify the performance of SciLitLLM.
>
> While we agree that domain-specific fine-tuning can provide performance gains in certain cases, **we respectfully argue that it is not always feasible for such models to outperform strong general-purpose models like GPT-4o, even within a specialized domain**. In scientific knowledge and literature understanding fields, for instance, Table 5 in SciKnowEval [2] — a benchmark for evaluating scientific knowledge in LLMs — shows that GPT-4o outperforms domain-specific models like Mol-Inst and ChemLLM by over 10%, where both models are instruction-tuned on extensive real-world and synthetic data.
>
> Similarly, Table 3 in SciRIFF [3], which specifically evaluates scientific literature understanding, demonstrates that both their 7B and 70B fine-tuned models underperform GPT-3.5T and GPT-4 by significant margins, with the 70B model trailing GPT-4 by over 10%.
>
> **In this context, SciLitLLM, despite being comparatively smaller in scale, achieves performance on par with GPT-4o, which we believe is a promising result**. Additionally, we would like to highlight the practical value of our work: there are real-world scenarios that require private, domain-specific models and cannot rely on proprietary general-purpose systems like GPT-4o. Addressing these needs is a key motivation behind our research.
>
> We hope these clarifications provide further perspective on the strengths of SciLitLLM, and we greatly appreciate your understanding and thoughtful evaluation!
>
> [2] SciKnowEval: Evaluating Multi-level Scientific Knowledge of Large Language Models
>
> [3] SciRIFF: A Resource to Enhance Language Model Instruction-Following over Scientific Literature

---

> > ### Author Response · Authors · 2024-11-21
> >
> > > Q3: Are there special reasons why we should group the results based on the 10B model size? (table 2/3). I think it's more reasonable to organize based on pre-trained only/with instruction tuning/with domain specific tuning?
> >
> > R3：Thank you for your suggestion.
> >
> > We grouped models by the 10B size threshold because models under 10B parameters can generally be deployed on consumer-grade GPUs (such as the Nvidia 3090 or 4090), whereas models exceeding 10B parameters typically require more specialized deep learning GPUs (such as the Nvidia A40 or A100). This distinction highlights deployment feasibility, which is an important consideration for many users.
> >
> > Following your recommendation, we have also revised Tables 2 and 3 in the updated paper to highlight models that include domain-specific training. Table 2 presents pretrained models, while Table 3 focuses on instruction-tuned models. We hope these adjustments make the results clearer and more accessible.
> >
> > > Q4: Also in this paper there is only fine-tuning results on the Qwen model family but not others. It would be interesting to compare the fine-tuning effects on llama or other model families.
> >
> > R4：Thank you for your comment!
> >
> > We agree that it would be valuable to explore the effects on a broader range of model families, such as LLaMA, and we plan to pursue this direction in future work. However, due to the considerable computational resources required for each pretraining run—approximately 2300 A100 GPU hours for a 7B model — we currently face limitations in our capacity to retrain additional models within the rebuttal period. We hope for your understanding regarding these constraints.
> >
> > > Q5: The author compared the performance with the SciTulu model, which is trained based on Llama-2 families. I don't think it's a fair comparison in table 3.
> >
> > R5: Thank you for your insightful comments. We appreciate your concern regarding the fairness of the comparison with SciTulu.
> >
> > We included SciTulu in Table 3, despite the model family difference, as it is currently the only official checkpoint available for scientific literature understanding.
> >
> > We agree that a fair comparison is essential to accurately assess the effectiveness of our SFT data, SciLitIns. SciTulu is based on the LLaMA-2 and is fine-tuned on both general-purpose data (Tulu v2 SFT mix) and domain-specific data (SciRIFF). For a clearer comparison, we also included a variant of SciLitLLM (Table 5, line 2) that uses the Qwen2.5 model fine-tuned on both general-purpose data (Infinity Instruct) and domain-specific data (SciRIFF). SciLitLLM outperforms this variant, which highlights the effectiveness of SciLitIns. We hope this provides clarity and addresses your concerns.

---

> > > ### Author Response · Authors · 2024-11-25
> > >
> > > Dear Reviewer 7iRT,
> > >
> > > Thank you for your detailed and valuable feedback on our submission! We have
> > > - clarified the limitations of our current PDF processing pipeline and future plans to improve it,
> > > - explained why SciLitLLM’s performance relative to GPT-4o aligns with domain-specific fine-tuning trends,
> > > - revised result grouping in Tables 2 and 3 for better clarity, and
> > > - justified the inclusion of SciTulu while addressing fairness concerns with additional comparisons.
> > >
> > > As the end of the discussion period approaches, we would like to ask if our responses were able to sufficiently address your concerns. If you have further questions, please let us know and we are eager to further address them!

---

> ### Author Response · Authors · 2024-12-02
> **Follow-Up: Have We Addressed Your Concerns?**
>
> Dear Reviewer **7iRT**,
>
> Thank you again for your thoughtful feedbacks on our submission, especially for advising us to 1) **clarifing the limitations of our current PDF processing**, 2) **explaining SciLitLLM’s performance relative to GPT-4o**, and 3) **revising results in Tables 2 and 3 for clarity**, 4) **justifing the inclusion of SciTulu for fairness**. These valuable suggestions have improved the clarity and quality of our work. We hope that these improvements will be taken into consideration.
>
> If our response has resolved your concerns on our paper, we will greatly appreciate it if you could re-evaluate our paper for a higher rating. We are also willing and ready to engage in discussions, if you have any further questions.
>
> Authors

---

> > ### Comment · Reviewer_7iRT · 2024-12-03
> > **Thanks for your clarifications!**
> >
> > Hi authors,
> >
> > I'd like to thank the authors for addressing the questions I brought up during the rebuttal period.
> >
> > I agree with the authors that achieving similar performance relative to GPT-4o models with relatively smaller models is a promising result; however I am not in particular excited considering it’s achieved requiring thousands of GPU hours of pre-training (brought up by the authors in the rebuttal).
> >
> > The authors clarification on the limitations of our current PDF processing does not resolve my concern: I think it’s OK to use PyPDF2 as a starting point but it should be important to quantify the degradation of the textual qualities of using such tools, for example, estimating the error rates of mixing footnotes with the main body of text, and to what extent the "Reformat & Grammar Correction" step can fix such issues. I do not see the detailed analysis in the rebuttal.
> >
> > I appreciate the author’s clarifying the results in the paper; however, I think the presentation of the main results (in table 3) can still be improved. For example, the author can simply instruction fine-tune Qwen2.5/Llama3 on the target dataset (needing only 32 hours for 7B models, line 354) and compare the performance with SciLitLLM-7B, in order to quantify the contribution of the pre-training stage.
> >
> > In addition, I suggest the authors consider other dimensions to assess the quality of the models – for example, one main issue with the Galactica models was that they do hallucinate a lot.

---

> > > ### Author Response · Authors · 2024-12-04
> > > **Thanks for your follow-up message**
> > >
> > > > Q1: I agree with the authors that achieving similar performance relative to GPT-4o models with relatively smaller models is a promising result; however I am not in particular excited considering it’s achieved requiring thousands of GPU hours of pre-training (brought up by the authors in the rebuttal).
> > >
> > > A1: Thank you for your thoughtful feedback. While we understand that the computational cost of pre-training our smaller model may temper enthusiasm, we would like to emphasize the broader value of our proposed pipeline. Specifically, this approach can be particularly advantageous for organizations or researchers with access to large, domain-specific corpora but limited expertise in domain adaptation techniques.
> > >
> > > Moreover, there are domains with minimal publicly available corpora where our framework, CPT, could prove highly beneficial. By leveraging proprietary or specialized datasets, CPT offers a viable path to achieving competitive performance in scenarios where adapting general-purpose models like GPT-4 is not feasible.
> > >
> > > We hope that the contributions of this work, including our resource, extend beyond the realm of scientific literature and inspire broader applications in private and niche domain modeling. We appreciate your feedback and will reflect this perspective in the revised version of the paper.
> > >
> > > > Q2: I think it’s OK to use PyPDF2 as a starting point but it should be important to quantify the degradation of the textual qualities of using such tools, for example, estimating the error rates of mixing footnotes with the main body of text, and to what extent the "Reformat & Grammar Correction" step can fix such issues. I do not see the detailed analysis in the rebuttal.
> > >
> > > A2: Thank you for your follow-up and additional constructive feedback. We acknowledge the importance of understanding how different PDF parsing tools/pipeline impact textual quality. However, quantifying degradation between various parsing tools and analyzing their error rates is beyond the scope of this work. Our main focus is on adapting LLMs for scientific literature understanding. While we utilize PyPDF2 as part of the pipeline, the nuances of parsing tool evaluation fall outside our primary objectives. That said, we agree this would be an interesting and valuable topic for future research. We hope this addresses your concern!
> > >
> > > > Q3: For example, the author can simply instruction fine-tune Qwen2.5/Llama3 on the target dataset (needing only 32 hours for 7B models, line 354) and compare the performance with SciLitLLM-7B, in order to quantify the contribution of the pre-training stage.
> > >
> > > A3: Thank you for your advice! We respectfully note that **we conducted an ablation study to evaluate the contribution of the Continued Pre-Training (CPT) stage** (Page 9, Table 4). We copy the table below:
> > >
> > > | Model                  | SciAssess | SciRIFF |
> > > |----------------------------|---------------|---------------|
> > > | Qwen2.5-Instruct           | 46.5          | 50.3          |
> > > | Qwen2.5-Base+SFT           | 49.8          | 57.0          |
> > > | Qwen2.5-Base+CPT+SFT (i.e. SciLitLLM)       | **51.7**      | **60.6**      |
> > >
> > > As shown in the table, incorporating CPT results in an extra 1.9% improvement on Sci-
> > > Assess and a 3.6% gain on SciRIFF. These results stress the unique role of CPT in pre-adapting the base model to scientific contexts.
> > >
> > > Due to time and resource constraints, we could not reproduce these experiments on Llama3, however we acknowledge it as a promising direction for future work to further confirm CPT's contributions.
> > >
> > > > Q4: I suggest the authors consider other dimensions to assess the quality of the models – for example, one main issue with the Galactica models was that they do hallucinate a lot.
> > >
> > >
> > > A4: Thank you for the suggestion! We acknowledge that assessing the model along dimensions such as hallucination is very important. However, we respectfully argue that this dimension is indirectly captured in our scientific literature understanding benchmarks, such as SciAssess and SciRIFF. If a model hallucinates and fails to ground its answers on the input documents, its outputs are likely to be incorrect, which would be reflected in lower benchmark performance.
> > >
> > > We would also like to clarify the scope and purpose of this work. **Our focus is on adapting general-purpose LLMs to excel in scientific literature understanding. While evaluating hallucination is an important topic, it falls outside the primary focus of this study.**
> > >
> > > We hope that these clarifications provide further clarity and context to our work.

---

### Official Review · Reviewer_TiCd · 2024-11-04

**Soundness:** 2
**Presentation:** 3
**Contribution:** 3
**Rating:** 6
**Confidence:** 4

**Summary:**

This paper introduces a method to improve scientific instruction following through post-training Qwen models. The main contribution is to collect science textbook data and improved SFT data mix. Their evaluation on a recent SciRiff dataset shows improvement.

**Strengths:**

- The paper introduces a strong pipeline in collecting textbook data and SFT data. This is aligned with most recent LM papers, showcasing the importance of data in the success of LM training.
- Improved result in science instruction following

**Weaknesses:**

The authors don't seem to be planning to release their textbook dataset. This raises the question of data contamination in evaluating the proposed model.

**Questions:**

Are you planning to release the textbook datasets?
Can you elaborate on the pipeline on what type of textbooks impact and improve task performance in SciRiff?
Can you provide contamination studies in textbook datasets and the test/eval cases?

**Details Of Ethics Concerns:**

I expect more explanation about the textbook datasets rather than mentioning "in-house" textbooks.

---

> ### Author Response · Authors · 2024-11-21
>
> > Q1: Are you planning to release the textbook datasets?
>
> R1: Thank you for your question. We have released a complete list of 73,000 textbooks, including textbook titles, authors and other meta-information (See the `books.xlsx` file in the updated supplementary materials).  Here are the first two rows of the textbook list:
>
> | Title                                                           | Author                                | Publisher               | Year | Pages | Language | Area                               | Sub-area           |
> |-----------------------------------------------------------------|---------------------------------------|-------------------------|------|-------|----------|------------------------------------|--------------------|
> | Mobile Satellite Communications: Principles and Trends         | Richharia Madhavendra                | Wiley                  | 2014 | 752   | English  | Engineering                        | Telecommunications |
> | The theory of island biogeography (Monographs in Population Biology) | Robert H. MacArthur, Edward O. Wilson | Princeton University Press | 1967 | 109   | English  | Biology and other natural sciences | Ecology            |
>
> Unfortunately, due to copyright restrictions on the electronic textbooks we use, we are unable to directly release the textbook files. However, with all our respect, we would like to highlight the **contribution we’ve made in terms of open-sourcing the data processing pipeline**.
>
> For researchers in academia or industry who wish to train private domain-specific models similar to ours, —especially those with access to large private domain copora but limited domain expertise — we hope our open-source pipeline can be a valuable resource. It allows for the construction of private domain models, not only limited to scientific literature.
>
> In fact, the need to build a domain-specific model for understanding scientific literature arose from our own practical challenges. We found that existing research and open-source projects did not directly address our demands, which motivated us to develop the data processing and model transfer pipeline described in this paper. While we are unable to release the textbooks themselves, we would like to highlight that by open-sourcing the complete data processing pipeline, we can provide helpful tools for both academic researchers and industrial practitioners working on similar scenarios in the future.
>
> We hope this clarifies our stance, and we appreciate your understanding.

---

> > ### Author Response · Authors · 2024-11-21
> >
> > > Q2: Can you provide contamination studies in textbook datasets and the test/eval cases?
> >
> > R2: Thank you for your insightful question! First, we want to note that prior literature consistently demonstrates that **data contamination in pre-training datasets has very small impact on evaluation performance**, for example:
> >
> > - The GPT-4 technical report states, `contamination overall has very little effect on the reported results` (Appendix C, Page 29) [1]. Notably, tasks like AP US History (73% contamination rate), AP World History (47%), and LSAT (39%) exhibit negligible differences in performance between the full evaluation set and clean evaluation set (excluding all contaminated data points).
> > - The PaLM authors compare PaLM's performance on both the full evaluation set and the clean evaluation set across 10 datasets with contamination rates ranging from 20% to 75%. They then conclude that `data contamination does not cause meaningful inflation of our reported results` (Section 8, Page 37) [2].
> >
> > To further examine the influence of contamination in the pre-training dataset, we conduct a detailed contamination analysis of our textbook and journal datasets using the contamination detection method from the GPT-4 technical report [1]: `For each evaluation example, we randomly select three substrings of 50 characters (or use the entire example if it’s less than 50 characters). A match is identified if any of the three sampled evaluation substrings is a substring of the processed training example.`
> >
> > The contamination rates are summarized below:
> >
> > | Eval Datasets  | In-house Textbooks (%) | In-house Journals (%) |
> > |-----------------|------------------------|------------------------|
> > | SciRIFF         | 1.1                   | 0.7                    |
> > | SciAssess       | 11.0                  | 1.1                    |
> >
> > The table shows that contamination rates are very low for SciRIFF but higher for SciAssess, especially in the textbook dataset. To assess whether contamination influences model performance, we evaluated SciLitLLM on SciAssess using both the full evaluation set and the clean evaluation set. The results are as follows:
> >
> > | Category     | Contamination Rates (%) | SciLitLLM-7B Full Set Accuracy (%) | SciLitLLM-7B Clean Set Accuracy (%) | SciLitLLM-14B Full Set Accuracy (%) | SciLitLLM-14B Clean Set Accuracy (%) |
> > |--------------|--------------------------|------------------------------------|-------------------------------------|----------------------------------|----------------------------------|
> > | Biology      | 12.2                    | 65.9                               | 66.3                                | 67.5                             | 67.6                             |
> > | Chemistry    | 13.3                    | 54.4                               | 55.8                                | 64.2                             | 65.2                             |
> > | Material     | 12.5                    | 52.5                               | 50.7                                | 61.8                             | 62.9                             |
> > | Medicine     | 8.7                     | 33.6                               | 34.2                                | 42.4                             | 43.9                             |
> > | Overall Avg  | 12                      | 51.6                               | 51.6                                | 59.0                             | 59.8                             |
> >
> > The results show **small differences in performance between the full and clean evaluation sets**. This aligns with prior findings, supporting the conclusion that data contamination does not significantly influence SciLitLLM performance.
> >
> > We hope this addresses your concern! Detailed experiments have been added to the Appendix G.
> >
> > Reference:
> >
> > [1] GPT-4 Technical Report. https://arxiv.org/abs/2303.08774
> >
> > [2] PaLM: Scaling Language Modeling with Pathways. https://arxiv.org/abs/2204.02311

---

> > > ### Author Response · Authors · 2024-11-21
> > >
> > > > Q3: Can you elaborate on the pipeline on what type of textbooks impact and improve task performance in SciRiff? I expect more explanation about the textbook datasets rather than mentioning "in-house" textbooks.
> > >
> > > R3: Thank you for your feedback! We would like to clarify the role of textbooks in enhancing SciRiff’s performance as follows. The performance of SciRiff relies on both the domain knowledge base and its ability to follow domain-specific instructions. Our choice of STEM textbooks as the corpus for continued pretraining plays a crucial role in effective domain knowledge injection. Specifically, we gathered a collection of **approximately 70,000 English STEM textbooks**, available to us with copyrights, for this purpose.
> > >
> > > However, due to the substantial computational resources required for each pretraining run (2300 A100 GPU hours for 7B model), conducting fine-grained ablation studies across different types of textbooks was beyond our current capacity. We hope you understand this limitation. Instead, we followed the methodology in [3], assessing text quality at the textual-piece level using an educational score metric. By filtering out low-quality passages, we improved the overall quality of the pretraining corpus, thereby strengthening the model’s domain knowledge and enhancing performance on SciRiff.
> > >
> > > Although we cannot directly provide the textbook dataset due to copyright restrictions, we will release the complete list of 73,000 textbooks. This list will allow researchers to identify the materials we used. Once researchers have obtained access to the PDF versions of these books, they will be able to use our open-source data processing and quality control code to prepare the data for continued pretraining, following our experimental setup.
> > >
> > > Reference:
> > >
> > > [3] Textbooks Are All You Need. https://arxiv.org/pdf/2306.11644

---

> > > > ### Author Response · Authors · 2024-11-25
> > > >
> > > > Dear Reviewer TiCd,
> > > >
> > > > Thank you for your thoughtful feedback on our submission, especially for advising us on
> > > > - **releasing the textbook dataset**,
> > > > - conducting **contamination studies**, and
> > > > - elaborating on the textbook pipeline’s role in SciRiff’s performance.
> > > >
> > > > These suggestions have improved the clarity and quality of our work.
> > > >
> > > > As the end of the discussion period approaches, we would like to ask if our responses were able to sufficiently address your concerns. If you have further questions, please let us know and we are eager to further address them!

---

### Author Response · Authors · 2024-11-21
**Common Response**

We sincerely thank all reviewers for their thoughtful and constructive feedback. We are encouraged by the positive comments on the motivation of our work (Reviewer fcV1), clarity of writing (Reviewer 7iRT), pipeline design (Reviewers TiCd, W9HG, and fcV1), model performance (Reviewers TiCd, W9HG, and fcV1), and dataset contributions (Reviewers 7iRT and W9HG). These insights have greatly helped us refine the paper, and we appreciate the opportunity to address the reviewers’ concerns. Below, we summarize the major updates made in response to the reviews:

- **CPT Dataset Release**: We have released the list of 73,000 textbooks used in constructing the CPT dataset. Unfortunately, due to copyright constraints, we cannot directly release the electronic textbook files. However, we woule like to highlight the value of our open-sourced data processing pipeline, which enables researchers in academia and industry to train private domain-specific models. This pipeline may be particularly helpful for those with access to large, domain-specific corpora but limited expertise in domain adaptation. We believe this resource extends beyond scientific literature and has broader applications for private domain modeling.

- **Contamination Studies**: To evaluate potential contamination effects, we tested SciLitLLM on SciAssess using both the full evaluation set and a clean evaluation set. Results indicate minimal differences in performance between the two sets.

- **Quality Filtering Threshold Experiment**: We performed a sensitivity analysis to examine how different quality filtering thresholds (the percentage of data filtered out from the pre-training corpus) impact model performance. The results showed that the current threshold of 25% yields slightly better performance compared to other tested thresholds.

We have made extensive efforts to address the reviewers’ main concerns, and the corresponding revisions are highlighted in orange throughout the manuscript. Additionally, detailed point-by-point responses to each reviewer’s comments are provided in the following sections.

Again, we appreciate reviewers' invaluable contributions toward improving the quality of this work.

---

### Meta-Review · Area_Chair_HLdK · 2024-12-22

**Metareview:**

This paper presents SciLitLLM, a language model tailored for scientific literature understanding, utilizing a hybrid approach that combines continual pre-training (CPT) with supervised fine-tuning (SFT). The proposed pipeline emphasizes constructing high-quality CPT datasets and generating diverse domain-specific instructions for SFT. The paper demonstrates competitive results on benchmarks like SciAssess and SciRIFF and introduces datasets and pipelines with potential applicability beyond scientific domains.

*Strengths*:
-Motivation and design: The paper provides a well-motivated approach addressing both domain knowledge enhancement and task-specific instruction alignment through its hybrid CPT+SFT strategy. Reviewers aknowledged the pipeline's clarity and the methodological rigor (fcV1, TiCd, and W9HG)
-Contributions: Reviewers appreciated the overall pipeline, including LLM-based data processing, format correction, and quality filtering mechanisms. The instruction synthesis method effectively addresses the challenge of limited scientific data. In addition the released models and data are valuable to the community.
-Empirical results: SciLitLLM achieves strong results compared with baselines and reviewers (fcV1 and W9HG) also noted its generalizability to other domains.

*Weaknesses*:

-Dataset processing: Reviewer 7iRT raised concerns about the use of PyPDF2 for document parsing. The authors acknowledged this limitation and clarified their choice was driven by computational constraints.
-Computational efficiency: Reviewer 7iRT questioned the computational cost-benefit ratio of achieving GPT-4o comparable performance
The authors provided context about the practical value of their approach for organizations needing private, domain-specific models.
-Comparative Analysis: Reviewers suggested additional comparisons with other model families and datasets. The authors conducted contamination studies and quality filtering threshold experiments during the rebuttal period to address these concerns.
-Closed nature of the textbook data: A major weakness is the closeness of the textbook data used in CPT. The authors do not have any plans to release such data, they might have legitimate copyright reasons though and they plan to release the list of the books.
-Some reviewers suggested comparisons (e.g., with Llama3). Such additional comparisons were acknowledged as valuable future work but were constrained by computational resources.

**Additional Comments On Reviewer Discussion:**

The rebuttal addressed some key reviewer concerns by improving clarity, releasing a list of textbooks used for pretraining, contamination studies showing minimal impact on evaluation performance, and validating the quality filtering threshold through empirical analysis. However, some other issues such as finetuning additional models, or change in the pdf pipeline isn't addressed. Such issues while difficult to address at the rebuttal time, should be considered in the next revision or camera ready.

---

### Decision · Program_Chairs · 2025-01-22

Accept (Poster)